

# A comparison of atmospheric $CO_2$ flux signals obtained from GEOS-Chem flux inversions constrained by in situ or GOSAT observations

Saroja M. Polavarapu[1], Feng Deng[2], Brendan Byrne[2], Dylan B. A. Jones[2], Michael Neish[1]

[1]Climate Research Division, Environment and Climate Change Canada, Toronto, Ontario, M3H 5T4, Canada
5 [2]Dept. of Physics, University of Toronto, Toronto, Ontario, M5S 1A7, Canada

*Correspondence to*: Saroja Polavarapu (saroja.polavarapu@canada.ca)

Key points:

1. The potential for GOSAT data to better resolve zonally asymmetric structures in the tropics year round and in the northern extratropics except during boreal winter is demonstrated.

10    2. In the lower troposphere, zonal asymmetries in the flux signal exceed that arising from meteorological uncertainties only in boreal summer, when in situ data constrain posterior fluxes.

3. The GEOS-Chem flux inversion constrained by in situ data better agrees (by 0.5 ppm) with independent observations on the global annual scale compared to the inversion constrained with GOSAT observations but the inversion with GOSAT data better captures the seasonal cycle of $CO_2$ at northern extratropical sites.

**Keywords:** data assimilation, carbon dioxide, GOSAT, $CO_2$ flux estimation, greenhouse gas sources and sinks, HIPPO, HIAPER Pole-to-Pole Observations, National ScienceFoundation, NSF, NSF/NCAR Gulfstream-V (GV).



**Abstract.** The $CO_2$ flux signal is defined as the difference of the four-dimensional $CO_2$ field obtained by integrating an atmospheric transport model with posterior fluxes and that obtained with prior fluxes. It is a function of both the model and the prior fluxes and it can provide insight into how posterior fluxes inform $CO_2$ distributions. Here, we use the GEOS-Chem

transport model constrained by either GOSAT or in situ observations to obtain two sets of posterior flux estimates in order to compare the flux signals obtained from the two different observing systems. Flux signals are also computed using two different models. The global flux signal in the troposphere primarily reflects the northern extratropics whereas the global flux signal in the stratosphere mainly reflects tropical contributions. While both observing systems constrain the global budget for 2010 equally well, stronger seasonal variations of the flux signal are obtained with GOSAT. Posterior $CO_2$

distributions obtained with in situ observations better agree with TCCON measurements over an 18-month time period, but GOSAT-informed posterior fluxes better constrain the seasonal cycle at northern extratropical sites. Zonal standard deviations of the flux signal exceed the minimal value (defined by uncertainty in meteorological analyses) through most of the year when GOSAT observations are used, but when in situ observations are used, the minimum value is exceeded only in boreal summer. This indicates a potential for flux estimates constrained by GOSAT data to retrieve spatial structures within

a zonal band throughout the year in the tropics and through most of the year in the northern extratropics. Verification of such spatial structures will require a dense network of independent observations.





## 1 Introduction

Flux inversion systems have become useful tools for understanding the global carbon budget as evidenced by their presence in Intergovernmental Panel on Climate Change (IPCC) reports (Ciais et al., 2013). However, even with the expansion of the near surface in situ network, limitations remain in the ability to retrieve regional-scale fluxes (Bruhwiler et al., 2011). Thus,

with the promise of retrieving fluxes with higher spatial resolution, the first satellite missions dedicated to greenhouse gas measurements from space were launched: the Greenhouse gas Observing SATellite (GOSAT) in 2009 (Kuze et al., 2009) and the Orbiting Carbon Observatory (OCO-2) (Crisp et al., 2015; 2017) in 2014. The expectation was that not only should space-based measurements of column integrated $CO_2$ offer better spatial coverage, but the column amount should be less sensitive to modelling errors associated with the Planetary Boundary layer (PBL) and its representativeness should better

correspond to that of coarse model grids (Keppel-Aleks, 2011). This occurs because mainly long range fluxes are seen in column data whereas both local and long-range flux signals are seen by surface in situ observations (Keppel-Aleks et al., 2011). Thus, space-based measurements of column integrated $CO_2$ offered the promise of alleviating some of the challenges associated with the assimilation of near surface in situ measurements in flux inversion systems. However, that promise has yet to be realized. Regional flux estimates have not been robust (e.g. Maksyutov et al., 2013; Basu et al., 2013; Chevallier et

al., 2014; Deng et al., 2014, Houweling et al., 2015) and they are sensitive to biases in satellite retrievals (Basu et al., 2013, Deng et al., 2014, Takagi et al., 2014). Retrieved fluxes from Europe are twice as large in GOSAT inversions compared to in situ inversions (Reuter et al., 2014; 2017), with many studies finding such increased sinks (Houweling et al., 2015; Feng et al., 2016). It has been suggested that the GOSAT-based inversions shift some uptake from North Africa to Europe which reduces the north-south gradient in $CO_2$ and reduces agreement with observations (Houweling et al., 2015). The issue may

also be due to the impact of nonlocal observations in flux inversion systems since biases in upstream $CO_2$ contribute 60-90% of the European sink (Feng et al., 2016). The point is that within the context of flux inversion systems, this new type of measurement poses new challenges. These challenges are related to aspects of the data specific to satellite column measurements. For example, biases arise from sampling only clear skies (Corbin et al., 2009; Parazoo et al., 2012) and from the seasonal variation of observational coverage (Liu et al., 2014; Byrne et al., 2017). At the same time, model transport

errors remain an issue for inversions using column measurements. Model errors in simulating boundary layer mixing are still important for assimilating column measurements (Lavaux and Davis, 2014), isentropic transport needs to be correctly modelled (Parazoo et al., 2012) and model biases in the high latitude upper troposphere can impact the north-south distribution of fluxes (Deng et al., 2015). Thus, it is important to get not only the low level vertical gradients correct in the transport model, but also the upper tropospheric and lower stratospheric distributions that the satellites are sensitive to.

Ultimately, the best network will combine surface and satellite measurements (Baker et al., 2006, Basu et al., 2013, Lavaux and Davis, 2014). The question is how to use the different types of observations to their strengths within a given data assimilation system.




The goal of this work is to improve understanding of how the different types of observations (near surface in situ observations versus space-based column measurements) inform model simulations of $CO_2$. To do this, we examine the flux signal which is defined as the difference between the $CO_2$ distribution obtained with retrieved fluxes and that obtained with prior fluxes. The flux signal reflects the impact of the observing system as manifested in the posterior fluxes convolved with

a time history of atmospheric transport for a given inversion system and is a function of both the model and prior fluxes. In particular, we compare the 3-D structure of the flux signal estimated from the in situ observing network and from GOSAT. Because satellite data are sensitive to the full column of $CO_2$ concentrations, accurate forward model simulations throughout the troposphere and lower stratosphere are needed in order to be able to correctly attribute model-data mismatch to upstream surface fluxes. Thus we focus on assessing posterior $CO_2$ distributions at various heights by comparing to observations. In

addition, the spatio-temporal evolution of the flux signal is examined through its global mean evolution, zonal mean structures, and zonal asymmetries. Two different tracer transport models, GEOS-Chem (http://geos-chem.org) and GEM-MACH-GHG (Polavarapu et al., 2016), are used to simulate the propagation of the flux signal. This allows an investigation of the sensitivity of our results to transport errors between the models. An additional benefit of using GEM-MACH-GHG is that we can obtain baseline estimates of uncertainty in the results by determining a "minimal" level of uncertainty in $CO_2$

distributions arising from imperfect knowledge of wind fields. The idea is that model-simulated $CO_2$ distributions have uncertainties due to a variety of sources (flux estimation errors, initial concentration field errors, model formulation errors, and wind field errors) but, at the very minimum, they will be impacted by wind field errors which can be substantial. For instance, Liu et al., (2011) estimate the uncertainty due to meteorology as 1.2–3.5 ppm at the surface and 0.8–1.8 ppm in a column mean $CO_2$ fields. Since GEM-MACH-GHG is a coupled weather and greenhouse gases (GHG) transport model, we

are able to determine uncertainties in our diagnostics that arise due to imperfections in meteorological analyses. Only when flux signal diagnostics exceed such minimum uncertainty levels do we find potential benefits of a given observing system.

The article is organized as follows. The experimental design is presented in section 2. Here the observations used for assimilation and verification, the transport models and the flux inversion system, are all described. Section 3 presents the results. First the posterior fluxes are compared (section 3.1) before the impact of posterior fluxes on $CO_2$ distributions is

examined (section 3.2). Diagnostics focus on comparison to independent observations of $CO_2$ and variations of the flux signal on global scales and in three zonal bands. Section 4 summarizes the results and considers their implications and generality.

## 2 The experimental design

In order to understand how the flux signal retrieved from assimilating atmospheric observations propagates into the vertical,

we must first perform some flux inversions. There are two sets of flux inversions performed with the GEOS-Chem model



and these are based on either the in situ observation network or on GOSAT column measurements. In order to assess the quality of the $CO_2$ distributions from the two observing systems, we compare posterior $CO_2$ distributions to independent measurements that contain some information about the vertical distribution of $CO_2$, namely, aircraft profiles from measurement campaigns, routine NOAA aircraft profiles and the ground-based column measurement network. Section 2.1

describes the observation systems used in the flux inversions as well as those used for validation of modelled $CO_2$ distributions. The models used are presented in section 2.2 while the flux signal and uncertainty estimate calculations are described in section 2.3.

### 2.1 The observations

The in situ observation network primarily consists of $CO_2$ mixing ratios measured by a nondispersive infrared absorption

technique applied to air samples collected in glass flasks at the NOAA ESRL Carbon Cycle Cooperative Global Air Sampling Network sites (Conway et al., 2011) and at the Environment and Climate Change Canada (ECCC) sampling sites. We use the same 72 NOAA sites and 6 ECCC sites that were used by Deng et al. (2014). **Figure 1** shows the approximate distribution of the insitu observations, as well as the validating observations (described below). Since observing stations may have missing data or may start or stop during the period of interest, the figure is only meant to provide a general idea of

the spatial distribution of the in situ observation network. While the coverage is global, the density of the stations is sparse, particularly in the tropics and southern hemisphere. On the other hand, the measurements are accurate, to better than 0.2 ppm (Tans and Thoning, 2016). The $CO_2$ measurements reflect the influence of local as well as remote sources (Keppel-Aleks et al., 2012; Byrne et al., 2017).

The satellite data used this paper are version b3.4 of the NASA Atmospheric $CO_2$ Observations from Space (ACOS)

GOSAT $XCO_2$ product, spanning July 2009 to December 2011, and have been bias corrected (Osterman et al., 2013). The ACOS retrievals employ an optimal estimation approach to infer atmospheric profile abundances of $CO_2$, from which $XCO_2$ is calculated. The details of the retrieval are described in O'Dell et al. (2012). Takagi et al. (2014) and Deng et al. (2014) showed that the biases of different versions of GOSAT products impact regional flux estimates but Deng et al. (2016) found that version b3.4 inferred fluxes result in $CO_2$ distributions that compare well to independent measurements. Hence, the

$XCO_2$ data used here are exactly those used in Deng et al. (2016). In addition, Deng et al. (2016) found that assimilating ocean glint measurements in addition to land nadir measurements results in generally improved agreement with independent observations and so both types of GOSAT data are used here also. **Figure 2** shows that, in contrast to the fixed locations of the ground-based in situ observations, satellite observations have a seasonal variation. In particular, in boreal summer when $CO_2$ uptake by the terrestrial biosphere in the Northern Hemisphere dominates the global $CO_2$ evolution, observations are

dense. In austral summer, the satellite's observational coverage shifts southward and the southern midlatitudes is observed





well. Throughout the year, ocean glint measurements observe the tropical oceans and improve the estimation of tropical fluxes (Deng et al., 2016).

The impact of the inversion results on $CO_2$ distributions are evaluated by comparing posterior $CO_2$ fields with atmospheric $CO_2$ observations from the Total Carbon Column Observing Network (TCCON) (http://tccon.ornl.gov/ ) (Wunch et al.,

2011). At the TCCON sites, solar-viewing ground-based Fourier transform spectrometers are used to measure high-resolution spectra (0.02 cm$^{-1}$) in the near infrared (3800–15,500 cm$^{-1}$), from which $XCO_2$ is retrieved. For the comparisons, we use observations from the current TCCON GGG2014 data set from 14 different sites (Blumenstock et al., 2014; Deutscher et al., 2014; Griffith et al., 2014a,b; Hase et al., 2014; Kivi et al., 2014; Notholt et al., 2014; Sherlock et al., 2014; Strong et al., 2014; Sussmann and Rettinger 2014; Warneke et al., 2014; Wennberg et al., 2014a,b). While total column

measurements can indicate the quality of modelled $CO_2$ simulations throughout the troposphere, they do not provide information on vertical distributions. For a more direct indication of model performance in the middle and upper troposphere, we also evaluate the inversions using aircraft data from the HIAPER Pole-to-Pole Observations (HIPPO) aircraft campaign (http://hippo.ornl.gov/) as well as NOAA aircraft profiles (Sweeney et al., 2015). Specifically, the 10 s averaged data from the HIPPO-3 campaign (Wofsy et al., 2012, 2011) are used for 24 March to 16 April 2010. The NOAA

aircraft profiles were limited to flights over Canada and the continental U.S. during 2010.

## 2.2 The models

### 2.2.1 The GEOS-Chem inversion system

The GEOS-Chem 4-dimensional variational (4D-Var) data assimilation system was used to estimate global regional $CO_2$ fluxes. The GEOS-Chem global 3-dimensional chemical transport model is driven by assimilated meteorological

observations from the Goddard Earth Observing System (GEOS-5) of the NASA Global Modeling Assimilation Office (GMAO). The model configuration is the same as that used in Deng et al. (2014). The horizontal resolution of the model is 4° x 5°, with 47 vertical levels extending from the surface to 0.01 hPa. The prior $CO_2$ fluxes, as described in Deng et al. (2014), include $CO_2$ fluxes from fossil fuel combustion and cement production from the Carbon Dioxide Information Analysis Center (CDIAC) (Andres et al., 2011), monthly mean shipping emissions of $CO_2$ from the International

Comprehensive Ocean-Atmosphere Data Set (ICOADS) (Corbett, 2004; Corbett and Koehler, 2003; Endresen et al., 2004; Endresen et al., 2007) , 3-D aviation $CO_2$ emissions (Friedl, 1997; Kim et al., 2007; Wilkerson et al., 2010), a climatology of monthly mean ocean-atmosphere $CO_2$ flux by Takahashi et al. (2009), biofuel $CO_2$ emission based on Yevich and Logan (2003), and monthly mean biomass burning $CO_2$ emissions from the Global Fire Emissions Database version 3 (GFEDv3) from van der Werf et al. (2010). The model includes 3-hourly Terrestrial ecosystem exchange from the Boreal Ecosystem

Productivity Simulator (BEPS) (Chen et al., 2012), which was driven by NCEP reanalysis data (Kalnay et al., 1996) and



remotely sensed leaf area index (LAI) (Deng et al., 2006). The annual terrestrial ecosystem exchange imposed in each grid box is neutral (Deng and Chen, 2011).

Two sets of inversions were performed using the two different observing networks for the 1 July 2009 to 30 June 2011 period (**Figure 3**). The first six months are treated as a spin up period and we mainly consider the estimated fluxes for
January 2010 – July 2011. The initial 3-D $CO_2$ mixing ratio fields were generated by running the model from January 1996 to December 2007 without assimilating any data, and then by assimilating surface $CO_2$ flask data from January 2008 to July 2010, following Deng et al. (2014). The optimized $CO_2$ mixing ratio field at 00:00 UTC on 1 July 2009, was used as the initial $CO_2$ field for the inversion analysis. As described in Deng et al. (2014), in assimilating the GOSAT data the model is transformed using the averaging kernels and prior $CO_2$ profiles from the $XCO_2$ retrievals. The assimilation did not account
for horizontal correlations in the observation and prior error covariance matrices. The uncertainties applied to the GOSAT and in situ data are the same as in Deng et al. (2016) and Deng et al. (2014), respectively. Specifically, the reported $XCO_2$ retrieval uncertainties were inflated by 1.90 over land and 1.02 over ocean. Uncertainties applied to in situ data were determined from model-observation statistics for each site. Prior flux uncertainties are 16%, of fossil fuel emissions and 38% of biomass burning per gridbox per month. An uncertainty of 44% is assumed for the ocean flux and a 22% uncertainty is
assigned to both the gross primary production and total ecosystem respiration per 3h per gridbox. Detailed explanations for these choices are found in Deng et al. (2014; 2016). Each set of inversions used a different assimilation window: 18 months for the insitu network but 12 months for the GOSAT network. This difference is necessitated by the different data densities. With the sparse insitu network, more time is needed to collect enough observations to determine upstream fluxes, therefore, we use, an 18-month window as in Deng et al. (2014). However, with the dense GOSAT observations (**Figure 2**), flux
signals have a greater chance of being observed quickly after injection into the atmosphere so a shorter window will suffice. Thus we used a 12-month assimilation window for the GOSAT inversion, as in Deng et al. (2016). Differences in the two inversion setups are inevitable because some parameters must necessarily differ (such as observation and representativeness error variances for the two measurement types). So, choosing exactly the same setup for both would force one system (and observation network) to be unfairly disadvantaged. Moreover, our intention is to examine the fluxes retrieved from what we
believe to be the "best" configuration for each.

### 2.2.2 The GEM-MACH-GHG model

GEM-MACH-GHG is a global, coupled weather and greenhouse gas prediction model with approximately 0.9° horizontal grid spacing and 80 vertical levels spanning the ground to the mesosphere (0.01 hPa). It is derived from the operational weather forecast model used for global and regional predictions by the Canadian Meteorological Centre and is described in
detail in Polavarapu et al. (2016). A semi-Lagrangian advection scheme is used for meteorology and constituent transport. For the latter, a global mass fixer was implemented. Convective transport of tracers through the deep convection scheme of



Kain and Fritsch (Kain and Fritsch, 1990; Kain, 2004) was also implemented. The posterior fluxes from the GEOS-Chem assimilation are inserted every model time step with 3 h updates. Note that posterior fluxes contain total fluxes from unoptimized as well as optimized fluxes. Since GEM-MACH-GHG does not yet have the ability to insert 3-dimensional emissions as GEOS-Chem does, the aircraft emissions were not inserted. This will lead to an underestimate in global $CO_2$ of

less than 0.1 ppm per year.

### 2.3 Estimating the flux signal and meteorological analysis errors

Once the flux estimates have been obtained, they are inserted into a forecast model to obtain posterior $CO_2$ distributions. Prior $CO_2$ distributions are also obtained by inserting prior fluxes into the same model and then the flux signal is determined by subtracting the prior $CO_2$ distribution from the posterior $CO_2$ distribution. Here we use GEOS-Chem as well as GEM-

MACH-GHG (Polavarapu et al., 2016) for this purpose. The advantage of using two models is that we can get a sense of the robustness of the results since the models will have different model errors. The disadvantage of using a different model (from that used for the flux inversions) to obtain the flux signal is that posterior fluxes contain an imprint of transport model errors from the model used for the flux inversion, so integrating these into another model will convolve the two transport models' errors (as seen in Polavarapu et al., 2016). If the two models' transport errors are fortuitously similar, then this

problem is avoided. However, this is unlikely to be the case for any two models on all time and spatial scales. Thus, we assess the ability of GEM-MACH-GHG to simulate $CO_2$ with fluxes derived from inversions performed with GEOS-Chem in order to identify where convolution of the two transport models' errors is evident.

By comparing $CO_2$ distributions from GEM-MACH-GHG obtained with posterior fluxes from GEOS-Chem with observations, we can assess the ability of this model to simulate $CO_2$ and search for instances of convolution of transport

model errors. **Figure 4** shows two-year time series of modelled and measured $CO_2$ at the NOAA or ECCC stations of Alert, Mauna Loa, Sable Island and South Pole. The two model simulations correspond to the two different flux estimates obtained with either in situ (red curves) or GOSAT (blue curves) observations. At Alert, which is far from $CO_2$ sources and sinks, a good comparison between the model simulation and measurements indicates a good ability of the model to transport the flux signal from the midlatitudes to the high latitudes on seasonal timescales. Indeed, **Figure 4** shows that the GEM-MACH-

GHG simulations agree rather well with observations at Alert using either set of fluxes although in boreal summer GOSAT-retrieved fluxes produce a better match than insitu-based fluxes. The better match with observations in boreal summer is consistent with the increased density of GOSAT observations in the northern hemisphere at that time (Byrne et al., 2017). The overestimation in boreal spring of both years with GOSAT-based fluxes was also seen in Deng et al. (2016) and suggests fortuitously similar transport by the two models to this location. The overall agreement of both GEM-MACH-GHG

simulations with Alert measurements is rather good especially considering the poorer agreement obtained with CarbonTracker 2013B (Peters et al., 2007, http://carbontracker.noaa.gov) fluxes in Polavarapu et al. (2016). This does not




mean that GEOS-Chem posterior fluxes are superior in any way to those of CT2013B, but rather that the transport errors of GEOS-Chem and GEM-MACH-GHG are fortuitously commensurate, at this location and time period. At Mauna Loa and Sable Island, which are far from sources but are also affected by synoptic scale variability, both model simulations compare well to measurements. At the South Pole, any differences in transport errors between the two models that accumulate over

long timescales are visible. Here a bias appears after 2 years of simulation with both sets of fluxes but the bias is smaller with in situ-based posterior fluxes. From the bias in the simulation with in situ fluxes we infer a mismatch of transport times to the southern hemisphere between the two models since GEOS-Chem simulations with the in situ-based fluxes match this station's time series well (not shown) and since a similar bias is also present between the two model simulations at other southern hemisphere stations (not shown). However, a positive bias of 0.5 ppm does appear when GOSAT-based posteriors

are used with GEOS-Chem (not shown). Thus the increased bias with GOSAT-data is seen by both models and is a separate issue from the convolution of transport errors.

The GEOS-Chem inversion was performed with a coarse 4°×5° resolution grid whereas GEM-MACH-GHG uses a much higher 0.9° resolution. So, the fact that the forward model simulations agree well with observations on synoptic time scales supports the contention of Agusti-Panareda et al. (2014) that the large scale gradients of $CO_2$ are captured in the retrieved

fluxes due to an adequate density of observations whereas the high resolution model captures and adds the correct synoptic scale variability. Overall, we conclude that GEM-MACH-GHG simulates $CO_2$ reasonably well with GEOS-Chem fluxes on a variety of timescales in the northern hemisphere, but there is mismatch of transport times to the southern hemisphere.

Polavarapu et al. (2016) showed that the existence of uncertainty in meteorological fields limits the spatial scales that can be depicted in $CO_2$ fields. Although it is only one of the many sources of error impacting $CO_2$ model distributions, it will

always be present and may be considered a minimum error level. To estimate this error, the forward simulations of GEM-MACH-GHG were repeated with perturbed meteorological fields and the difference in $CO_2$ defines this inescapable error. To perturb the meteorological fields, Polavarapu et al. (2016) simply computed the difference between the meteorological analyses valid at the required time and those valid 6h prior to the required time, and then removed the diurnal signal from this perturbation. Here we improve on the methodology by using actual realizations of analysis error from our operational

ensemble Kalman Filter (EnKF) system (Houtekamer et al., 2014), which is used to determine meteorological forecast uncertainty on the medium range. Because the ensemble members were not available in the archives of the Canadian Meteorological Centre for the period of study here, we use analysis error estimates from a different year. The supplemental material describes how the perturbations were computed and demonstrates that the method used to estimate analysis errors is considerably better that used in Polavarapu et al. (2016) despite some unavoidable approximations. In this work,

meteorological fields perturbed by EnKF-derived meteorological analysis errors will be used to define minimum error levels in the diagnostics of sections 3.2.4 and 3.2.5.





## 3 Results

The two sets of posterior fluxes that will be used to study the $CO_2$ flux signals are described in section 3.1 before considering the vertical propagation of the flux signal in section 3.2. While some of the figures below include results from both models, others show those from a single model. In such cases, results from the GEOS-Chem model are shown, while corresponding

figures obtained with GEM-MACH-GHG are relegated to the supplemental information. This choice was made because GEOS-Chem was used in the flux inversions, so posterior $CO_2$ distributions with GEOS-Chem are obtained with consistent model errors while posterior distributions obtained with GEM-MACH-GHG will convolve the transport errors from the two models. However, despite this convolution of errors, consistent patterns emerge with both models, lending greater confidence in the robustness of results in the face of transport error.

**3.1 Posterior flux estimates**

The global total flux estimates for 2010 obtained from the two observation networks studied here are 5.01 Pg C (in situ) and 4.95 Pg C (GOSAT). Here positive values indicate fluxes from the Earth's surface into the atmosphere. The actual annual growth rate for 2010 from Conway and Tans (2012) is 2.41±0.06 ppm or 5.12±0.13 Pg C (using a conversion factor of 2.124 Pg C ppm$^{-1}$). The general agreement of both sets of posterior fluxes with the 2010 annual total flux suggests that both

inversions are sufficiently well configured.

While the global annual totals for 2010 are similar with the two different observation networks, the spatial distributions of the fluxes differ. **Figure 5** shows the spatial distributions of the annual flux for 2010 for the 11 TransCom (Gurney et al., 2003) land regions. The prior and the in situ-based posterior fluxes are similar to those shown in Figure 4 of Deng et al. (2014) while the GOSAT-based posterior fluxes are similar to those presented in Figure 8 of Deng et al. (2016). As in Deng

et al. (2014), **Figure 5** reveals that in situ data result in more uptake in the Americas whereas fluxes retrieved from GOSAT data put more uptake in Eurasia. As noted in this Introduction, this increase in European uptake with GOSAT data was also seen by Reuter et al. (2014) and Houweling et al. (2015).

Both sets of posterior fluxes have the same sign for most regions except North and South Africa and Australia. In the north-south direction, in situ fluxes produce more uptake in the three tropical regions compared to GOSAT-derived fluxes, while

the latter have relatively more uptake in temperate and boreal Eurasia. This was also seen in Houweling et al. (2015). This difference in north-south distributions of fluxes is more readily evident in **Figure S6** which shows the temporal variation of the fluxes accumulated over three large latitudinal bands: the northern extratropics, the tropic and the southern extratropics. (Here the dividing latitude between the tropics and extratropics is taken as 19.47° or $\sin^{-1}$ (1/3) because it results in exactly equal areas for all three regions. This advantage is exploited later to interpret the diagnostics of section 3.2.) **Figure S6a**

reveals that both sets of fluxes are generally similar on the global scale with two exceptions: (1) the peak boreal summer



uptake occurs in June with GOSAT data, but in July with in situ data, and (2) GOSAT data produces larger outgassing of $CO_2$ in October and November. The larger outgassing with GOSAT data in boreal autumn is due to larger contributions from both the northern extratopics (**Figure S6b**) and the tropics (**Figure S6c**). The larger global uptake in June with GOSAT data is due to the northern extratropics (**Figure S6b**). In the southern extratropics, GOSAT generally results in

more uptake than in situ data but the magnitude of the uptake and the difference between the two posterior fluxes is small (**Figure S6d**). The temporal variation of the flux is further broken down into the 11 TransCom land regions in **Figure 6**. The greater outgassing in boreal autumn with GOSAT data is seen in the Boreal and Temperate North American regions, as it was in Deng et al. (2014, their Figure 5). In temperate North America, the $CO_2$ source is greater throughout the boreal winter months but in boreal North America, the GOSAT fluxes are larger only in October and November, as seen in **Figure**

**S6a-b**. Also, as noted by Deng et al. (2014), the boreal Eurasian sources in boreal winter are close to their prior values due to the dearth of observations there combined with low prior flux values (implying low prior flux uncertainty). In the tropics, there is little agreement between the two sets of fluxes. In Australia, the in situ flux stays close to the prior value but GOSAT-derived fluxes tend to decrease the prior through most of the year. This adjustment with GOSAT data occurs because the measurements observe the southern hemisphere well through ocean glint measurements (Byrne et al., 2017)

whereas in situ observations are sparse in that region.

   In summary, the posterior fluxes produced here bear similarities to those produced by other inversion systems constrained by similar observation sets, and are consistent with the range of results of the multi-inversion intercomparison of Houweling et al. (2015). Thus the two sets of posterior fluxes may be considered as reasonable examples representative of the two observing systems. Furthermore, the results obtained here should be relevant to other flux inversion systems.

**3.2 Vertical propagation of the flux signal**

Given the two sets of posterior fluxes, we now consider how they inform atmospheric $CO_2$ distributions. Although column measurements contain information about $CO_2$ concentrations throughout the depth of the troposphere, ultimately, in a flux inversion, this information is used to update a surface flux. It is unclear how this updated surface flux signal then propagates vertically to inform the middle and upper troposphere. Intuitively, one might expect the assimilation of column

measurements to result in better $CO_2$ depictions in the middle and upper troposphere. However, as will be shown, this is not necessarily the case.

**3.2.1 Zonal mean patterns**

The flux signal was computed for both sets of posterior fluxes resulting in four sets of 4-dimensional $CO_2$ fields—two sets for each model. To encapsulate the vertical motion, zonal mean fields were computed. The GEOS-Chem fields are

animated in **Figure S7** and snapshots from the animation, taken every 3 months from 1 October 2009 to 1 July 2011 are





shown in **Figure 7**. Qualitatively similar results are obtained with GEM-MACH-GHG (see supplementary material). Immediately obvious from **Figure 7** is that the flux signal is largely negative for both experiments at all times. This occurs because the prior flux has a terrestrial component that produces an annually balanced biospheric flux. Thus, the fact that the terrestrial biosphere annually takes up approximately 30% of the anthropogenic emissions entering the atmosphere (Le

Quéré et al., 2015) is not assumed by the prior fluxes. This is done intentionally because of the desire that observations determine the existence and amount of uptake by the terrestrial biosphere. Here, the impact of using annually balanced biospheric fluxes and ocean prior fluxes from Takahashi (2009) that only account for 1.4 of the expected 2.5 Pg C per year is that the prior $CO_2$ distribution has a continually increasing global total relative to the actual increase. Then, once the flux inversion is performed and the fluxes are pulled toward realistic values, the posterior distributions reduce the overestimated

$CO_2$. Thus the difference between the posterior and the prior $CO_2$ distributions is always negative, in a global sense—hence the overwhelming negative values seen in **Figure 7**.

 Comparing the distributions obtained with the two observing systems reveals some clear patterns. In October 2009 (which is still in the spin up period), the patterns are similar except that the GOSAT data produce a smaller flux signal in the tropics. This is even more evident by January 2010, where the GOSAT-derived flux signal has higher $CO_2$ in the northern

hemisphere as well. At this time, there is a clear difference in the vertical gradient of the flux signal between the tropics and northern extratropics, with GOSAT data producing reduced meridional gradients. This was also seen by the inversion systems in Houweling et al. (2015, their Fig. 8), but the reduced gradient was not supported by independent measurements. By April 2010, the in situ data are continuing to reduce $CO_2$ in the northern hemisphere and tropics, while the GOSAT data seem to not have much impact. On the other hand, in July 2010, GOSAT data produce a large negative flux signal in the

northern hemisphere when the satellite observes this region well (**Figure 2**). However, the tropical upper troposphere retains a stronger flux signal with the in situ data. In the second year of simulation, these patterns are repeated as the troposphere slowly adjusts to more realistic global mean values resulting from the observationally constrained terrestrial biospheric uptake. Specifically, October 2010 sees similar patterns for the two simulations in the northern hemisphere and tropics while January 2011 reveals larger $CO_2$ throughout the troposphere in GOSAT-based simulations. April 2011 again sees a greater

reduction in $CO_2$ throughout the tropics with in situ data so that the GOSAT-based flux signal is relatively lower throughout the troposphere. Finally, by the end of June 2011, the large flux signal obtained with GOSAT data is seen once again in the northern hemisphere while in situ data retains a large flux signal in the tropical troposphere. When these patterns are animated (**Figure S7**), it appears that the in situ data provide a constant injection of information from northern hemispheric fluxes which is transported upward and equatorward to inform the tropical middle and upper troposphere. GOSAT data

provide large updates to fluxes in boreal summer in the northern hemisphere, but when boreal autumn comes and the satellite tracks shift southward, the flux signal diminishes. In boreal winter, GOSAT observes the southern hemisphere well, but the northern hemisphere dominates the global $CO_2$ seasonal variation (Keeling, 1960) and so GOSAT misses the northern





hemisphere emissions and the flux signal diminishes in this hemisphere with subsequent missing transport of the flux signal to the tropics. In fact, Houweling et al. (2015) argue that this seasonal variation of GOSAT data coverage plays a role in amplifying the European sink. This difference in the seasonality between inversions with in situ and GOSAT data is also consistent with the results of Byrne et al. (2017). Although both simulations only adjust surface fluxes, the in situ-based

posterior fluxes constantly inform the northern hemisphere and the adjusted $CO_2$ patterns are transported upward to the tropics. This transport of information relies on the accuracy of the model's transport and hence may not be correct. Transport error has long been known to be a major source of error in flux inversion systems (e.g., Chevallier et al., 2014; Chevallier et al., 2010; Houweling et al., 2010; Law et al., 1996). Thus, to see which of the two posterior fluxes better depicts the middle and upper troposphere, we compare to independent measurements in the next subsection.

**3.2.2 Comparison to observations**

Total column measurements from the TCCON provide indirect information about $CO_2$ concentrations throughout the troposphere. The dominant feature seen in seasonally-aggregated comparisons of modelled $CO_2$ to TCCON is the larger bias resulting from GOSAT-based fluxes (**Figure 8, 9).** At all stations, except Eureka, a difference of about 0.5 ppm between the biases of the two simulations is present, with in situ data providing a closer fit to the measurements (**Figure 8**).

However, if we look beyond this time-mean bias (by subtracting it out), GOSAT-based fluxes are seen to better define the seasonal cycle (**Table S1**) at most northern extratropical sites. Visually, this means the black curves are generally flatter than the red curves in **Figure 9**. At most of the northern sites (Bialystok, Garmisch, Izana, Karslruhe, Lamont, Orleans, Park Falls, Sodankyla) the seasonal variation of the statistics obtained with GOSAT is better since the means of absolute anomalies are lower than those obtained with in situ data (compare columns 6 and 7 in **Table S1**). (The results from Eureka

seem anomalous relative to other TCCON sites so this is a topic currently under investigation by Kimberly Strong. Explanations under consideration include sampling issues, site-to-site differences, model transport errors and unknown issues with the data.) The improved ability of inversions constrained by GOSAT data to capture the seasonal cycle was also found by previous analyses (e.g., Deng et al., 2014; Liu et al., 2014; and Reuter et al., 2014). Butz et al. (2011) and Lindqvist et al. (2015) showed that GOSAT/ACOS data alone can match the seasonal cycle at TCCON locations (typically

within 1 ppm in the Lindqvist study). In addition to better capturing the seasonal cycle, the GOSAT-based simulations result in lower mean residuals at many of the northern hemisphere sites in June, July and August (**Figures 9**). Improved agreement with independent observations using posterior fluxes from the GOSAT inversion relative to the in-situ inversion during boreal summer was also found by Basu et al. (2013), Deng et al. (2014) , and Reuter et al. (2014) and suggests that the summer drawdown in the in-situ inversions is too weak over the northern extratropics. Overall, however, the posterior

fluxes obtained with in situ observations provide better agreement with TCCON overall since 61 of the 76 (80%) comparisons favour the simulation based on in situ data (**Figure 9**). The standard deviations are rather similar for the two simulations and are frequently smaller than the means (**Figure 9**).





Comparisons to TCCON obtained with GEM-MACH-GHG posterior $CO_2$ distributions are found in **Figures S10 and S11**. The same conclusions hold: there is an overall larger mean mismatch with TCCON when GOSAT-based posteriors are used (**Figure S10**) but the seasonal cycle is better captured at most northern extratropical sites (**Table S1**) and the agreement in boreal summer is better at many northern extratropical sites (**Figure S11**). Additionally, a convolution of the two models'

transport errors is evident in **Figure S10** in that a larger bias with TCCON at southern hemisphere sites is seen with GEM-MACH-GHG compared to that obtained with GEOS-Chem (**Figure 8**). This bias was also seen in **Figure 3** (South Pole station) and arises due to differing transport times from the tropics to the southern hemisphere in the two models. GEOS-Chem transports $CO_2$ more rapidly to the southern hemisphere and its posterior fluxes reflect this rapid transport (see animation in **Figure S8,** especially July-August 2010). When inserted into GEM-MACH-GHG, the fluxes obtained

assuming a fast transport to the southern hemisphere result in a too-slow departure from the prior $CO_2$ distribution and a larger bias with respect to observations. However, because GEM-MACH-GHG is disadvantaged by the convolution of transport errors, these results do not identify which model's interhemispheric transport to the south is more realistic. As a weather and environmental forecast model, knowledge of the age-of-air for GEM-MACH is not essential for its time scales of interest so this work identifies a need to better characterize interhemispheric transport with the GHG version of this

model. At the same time, this work shows little evidence for the convolution of transport errors on shorter time scales or in the northern hemisphere (as was seen when GEM-MACH-GHG used CT2013b fluxes in Polavarapu et al., 2016). Moreover, despite the existence of some convolution of transport errors, conclusions regarding the agreement with independent measurements hold for both models, increasing confidence in the robustness of results in the face of model errors.

A more direct assessment of middle and upper tropospheric $CO_2$ distributions is obtained by comparing to aircraft profiles. Comparisons of both GEOS-Chem simulations to measurements from the HIPPO-3 campaign in 24 March to 16 April 2010 are shown in **Figure 10**. The results are aggregated by latitude and vertical bands. The in situ-based posterior fluxes result in lower mean differences from measurements in the middle to upper troposphere (panel c) and the lower stratosphere (panel d) in the northern extratropics. However, the GOSAT-based posterior fluxes generally agree better with measurements in

the southern extratropics at all heights. Similar results are also obtained with GEM-MACH-GHG (**Figure S12**) in the northern extratropics but in the southern extratropics, in situ fluxes better match observations because of convolution of transport errors which leads to increased $CO_2$ in the southern hemisphere for all fluxes. Note that in the stratosphere for both comparisons with HIPPO-3 (**Figures 10d** and **S12d**), the mean mismatch exceeds the standard deviation. This means that both model simulations are biased in the stratosphere as was seen in Deng et al. (2015). Such a bias can adversely affect flux

estimates in the northern hemisphere (Deng et al. 2015). Comparing **Figures 10** and **S12** (panels c and d) reveals that GEM-MACH-GHG has better agreement with HIPPO-3 in the middle to upper troposphere and in the stratosphere. This makes sense given the finer vertical and horizontal resolution of GEM-MACH-GHG and is expected from the results of Deng et al.





(2015, their Figures 11-12). The number of realizations used in each comparison in **Figures 10** and **S12** ranges from 94 to 2570 and the differences in the mean values of the two experiments are significant at the 90% level. Thus, overall, we conclude that the middle and upper tropospheric distributions of $CO_2$ are better in the northern hemisphere in boreal spring 2010 when posterior fluxes use in situ data rather than GOSAT column measurements.

Since measurement campaigns occur only in select time windows (HIPPO-3 was in March-April 2010), we also consider the more routine NOAA aircraft profile measurements from continental U.S. and Canadian sites in **Figure 11**. The observations are from ObsPack2013 (Masarie et al., 2014). As in Agusti-Panareda et al. (2014), mean model profiles at the nearest grid point and time step to the observation locations and times are averaged over a season. Observed values are binned into 1 km layers and compared to model values at mid-layer. Hourly GEOS-Chem fields are used. When the entire 2010 year is

considered, the bias throughout the troposphere with respect to the aircraft profiles is much smaller with in situ-based posterior fluxes (**Figure S13**) for both models. However, the results are more variable if broken down by season. In boreal winter, in situ data produce better agreement with NOAA aircraft near the surface but from 2-6 km GOSAT data give a better result (**Figure 11**). This variation in fit is related to the fact that vertical profiles from GEOS-Chem have stronger than observed gradients in the lowest 1-2 km. GEM-MACH-GHG profiles better match observed gradients (**Figure S14**) and

GEM-MACH-GHG profiles consistently favour the same simulation in both height ranges. In Dec.-Feb. 2010, the in situ-based simulation better matches observations although it is partly in the spin up period, whereas in Dec.-Feb. 2011, the GOSAT-based simulation better matches mean NOAA aircraft profiles at all heights (**Figure S14**). In boreal spring (**Figures 11 and S14**) in situ data produce better agreement not just near the surface, but at all heights. In boreal summer, GOSAT data result in much better agreement from 1-3 km but from 3-6 km there is little difference between the two

simulations. However, in boreal autumn, GOSAT data achieves a better match from 2-6 km, whereas in situ data has a better match near the surface (**Figures 11**). As in boreal spring, incorrect vertical gradients obtained with GEOS-Chem are likely playing a role in the inconsistent results since GEM-MACH-GHG's vertical gradient is closer to that observed and it favors the GOSAT-based simulations at all heights (**Figure S14**). Overall, from 3-6 km, simulations with both posteriors produce similar model profiles in boreal summer and fall, but in boreal winter and spring there is a difference between the

two, with in situ data producing lower $CO_2$. The lower $CO_2$ values obtained with in situ data agree better with aircraft data in boreal spring, but not in boreal winter. From 1-2 km, in situ data better match aircraft data in boreal spring while GOSAT achieves the better match in boreal summer, for both models. These results once again confirm that in boreal summer when GOSAT views and samples the northern hemisphere well, the estimated fluxes are improved in the lower troposphere.

     In summary, the results that are consistent are as follows. (1) Despite the reliance on faithful model transport, in situ-based

posterior fluxes produce $CO_2$ distributions that better agree with independent observations of the middle troposphere in the northern hemisphere in boreal spring. This may partly be due to the propagation of the near surface improvements obtained





in boreal winter. (2) GOSAT-based posterior fluxes consistently achieve better agreement with independent observations in the northern hemisphere in boreal summer and in the middle to upper troposphere in boreal winter.

### 3.2.3 Adjoint Sensitivity

**Figures 7** and **S9** as well as animations **S7** and **S8** imply a propagation of flux signal from the northern midlatitude lower
troposphere to the tropical middle and upper troposphere with in situ-based posterior fluxes. The question of whether this is realistic or not was the subject of the previous subsection where model simulations were compared to independent observations. Here we consider whether such transport (realistic or otherwise) has implications for flux inversions. In other words, can $CO_2$ from the northern extratropics influence the $CO_2$ in the tropical upper troposphere a few months later? To see whether this occurs in the flux inversion system, we compute the sensitivity of $CO_2$ at one point in time with respect to
the $CO_2$ state at an earlier point in time using the adjoint of GEOS-Chem (Henze et al., 2007). While Byrne et al. (2017) utilize the adjoint sensitivity with respect to surface fluxes, here we need to consider the entire $CO_2$ state in order to see vertical transport of information. The extension of the adjoint calculation needed to produce sensitivity to the $CO_2$ state is described in Appendix A, and **Figure 12** shows the sensitivity of the $CO_2$ field on 1 February 2010 to earlier states, at one month intervals. Each panel shows a snapshot of the zonally averaged sensitivity field. In February 2010, the sensitivity is
initialized to a uniform value within a mask from 20°S-20°N and 500-250 hPa. Proceeding backward in time, this field is sensitive to the $CO_2$ field throughout the depth of the tropics in January 2010 with a hint of sensitivity beyond the tropics in the stratosphere. By November 2009, this stratospheric influence is more evident and by October 2009, extratropical tropospheric influence is also evident. By September 2009, the sensitivity is largest in the northern and southern extratropics. Tracing the pattern in the northern hemisphere forward in time through the panels reveals upward and
equatorward propagation of the signal. Thus the $CO_2$ field in the northern tropics in the upper troposphere in boreal spring is sensitive to $CO_2$ in the northern midlatitude lower troposphere on September 1. In other words, observations near the surface at northern midlatitudes on September 1 can potentially impact $CO_2$ fields in the tropical upper troposphere, 3 to 6 months later. Because the adjoint calculation only reveals patterns without a magnitude (since the actual influence of an observation on $CO_2$ estimates also involves error covariances of observations and propagated prior flux errors), only a potential influence
can be revealed in **Figure 12**. However, this potential influence is sufficient to demonstrate the atmospheric transport from the northern midlatitude lower troposphere to the tropical upper troposphere on the timescale of several months. This figure then supports the notion that observations of the northern midlatitudes combined with model transport can influence (rightly or wrongly) $CO_2$ distributions downstream in the middle and upper tropical troposphere.

### 3.2.4 Global mean flux signals

The flux signal modifies $CO_2$ fields locally but, eventually, gradients get diffused by atmospheric turbulence and only the impact on the background $CO_2$ field is retained. Thus, looking at the zonal or global mean flux signal reveals long time scale



information retained from flux adjustments after redistribution and dispersion by model transport. How long does it take for a flux signal to modify the background $CO_2$ state? Deng et al. (2014) show that transit times of regional fluxes to the middle troposphere further downstream are shorter than two months and flux signals have dispersed to the background within 3 months (see their Figure 15). Similarly, Liu et al. (2015) found that column measurements are unable to distinguish the

locality of fluxes older than three months. **Figure 13** shows the globally averaged zonal mean flux signal for both models and both observing systems at selected model levels in the lower troposphere (panel a), the middle troposphere (panel b), the upper troposphere (panel c), the lower stratosphere (panel d), middle stratosphere (panel e) and the upper stratosphere (panel f). (It was possible to find similar model levels in terms of approximate pressure for the six representative pressures for the two models by assuming a 1000 hPa reference for each vertical coordinate. These are listed in **Table 1**.) From Deng et al.

(2014) and Liu et al. (2015) we conclude that the time scales reflected in **Figure 13** are seasonal and longer time scales. The evolution of global $CO_2$ when forced by the prior flux is missing a trend due to the assumption of a balanced biosphere so the prior $CO_2$ fields drift from a realistic global mean, increasingly overestimating it. Since the posterior $CO_2$ fields are constrained by observations to resemble the actual atmospheric budget evolution, our global flux signal increases with time as the trend error accumulates (**Figure 13** black curves). (Here the posterior fields are subtracted from the prior fields to

give positive values, for convenience.) **Figure 13** shows that the global flux signal increases not only for the atmosphere as a whole but also at all heights (except the upper stratosphere). In addition, for the GOSAT-based flux signal, there is a large seasonal variation on top of the linear trend which has largest amplitude near the surface.

**Figure 13** also shows that despite the differing transport errors, the global flux signals are very similar for the two models. The largest differences occur in the upper troposphere and lower stratosphere (UTLS) regions (panels c and d). As noted by

Deng et al. (2015), the GEOS-Chem $CO_2$ simulation at a resolution of 4° x 5° is biased in the UTLS. Stanevich et al. (manuscript in preparation) found a similar bias in the coarse resolution $CH_4$ simulation in GEOS-Chem, which they attributed to excessive mixing across the tropopause at the 4° x 5° resolution. Compared to the flux signal obtained with in situ data, the flux signal derived from GOSAT data diminishes in boreal winter and spring throughout the troposphere. Recall that in boreal spring, in situ data provided the better match of $CO_2$ distributions to NOAA aircraft in the lower

troposphere (**Figures 11, S14**). In the stratosphere, the overall signal is smaller with GOSAT data, but there is little seasonality to the signal for either experiment (**Figure 13d-f**). Because the flux signal reflects the departure of the posterior from the prior $CO_2$ field, it is not clear whether a large or small seasonal variation should be expected. However, comparisons of posterior fields to measurements in section 3.2.2 revealed that the posterior $CO_2$ fields derived from in situ data have an approximately 0.5 ppm lower bias relative to TCCON at all sites except Eureka. They also agree better with

NOAA aircraft and HIPPO-3 in the middle and upper troposphere in boreal spring and with NOAA aircraft at all heights when annual mean profiles are considered. This suggests that the larger signal seen in boreal spring with in situ data may be realistic. In boreal winter, near the surface, the $CO_2$ fields obtained from in situ posteriors agree better with NOAA aircraft




profiles, but those based on the GOSAT posteriors yield better matches from 2-6 km. However, the NOAA aircraft data used corresponds to North America, whereas **Figure 13** illustrates global diagnostics while the overall TCCON comparison (**Figure 8**) suggests in situ distributions are more realistic. Thus, it is not entirely clear whether the larger signal seen in boreal winter with in situ data is more realistic than the lower one obtained with GOSAT data. What is clear is that flux

inversions that assimilate GOSAT data produce posterior distributions that are less consistent with observations in global, annual statistics than flux inversions using in situ data. In addition, the GOSAT-informed flux signal has much stronger seasonal variations than the in situ-based flux signal. Thus, sub-annual variations in the global mean $CO_2$ are sensitive to the observing system used. However, this sensitivity also depends on the choice of prior fluxes since, for example, a prior flux with reduced bias in boreal summer would reduce this effect.

How much can we trust the global flux signal? The model transport of flux adjustments is not perfect and a major component of the transport model uncertainty is due to wind field errors (Liu et al., 2011). We can use the coupled meteorology and greenhouse gas transport model to identify the error due to wind field uncertainty on flux signals by simply repeating each simulation with perturbed meteorological fields (as described in the Supplemental material). The difference in posterior $CO_2$ distributions obtained with the control and perturbed meteorology defines this uncertainty. This uncertainty

is plotted in **Figure 13** but is not evident because the curves are near zero. This is not surprising because the global mean atmospheric $CO_2$ is independent of transport. It is the spatial distribution of $CO_2$ that is affected by atmospheric transport (as will be demonstrated shortly). However, by considering the global mean at various heights, there was the possibility that an influence of errors in atmospheric transport might be seen at some vertical levels.

**Figure 14** shows how the tropics and extratropics contribute to the global flux signal based on GOSAT data and computed

with the GEOS-Chem model. As noted earlier, the dividing latitude between the tropics and extratropics was chosen so that the three zonal bands have equal areas. Because the zonal bands have equal areas, we multiply the zonal contributions depicted in **Figure 14** by a factor of three, which means that each regional total (red, blue or green curves) can be compared to the global total (black curves). For example, **Figure 14a** reveals that in the lower troposphere, the dominant contribution to the global flux signal comes from the northern extratropics where there is a large seasonal variation due to the seasonality

of observational coverage (**Figure 2**) in addition to the seasonality in the fluxes. This is also true for the middle troposphere (**Figure 14b**). However, in the lower and middle stratosphere, the tropics dominate the global flux signal (**Figure 14d-e**). The upper stratosphere is not much influenced by flux adjustments (**Figure 14f**) on the two-year time frame. Since the northern extratropics dominates the global flux signal, the concern of Houweling et al. (2015) that the excellent observational coverage of this region by GOSAT in boreal summer combined with the poorer coverage in boreal winter has

implications on flux inversions seems warranted. **Figure S15** shows that these patterns also occur for flux signals derived from assimilating in situ observations but the seasonal variation of the flux signal is greatly reduced. The flux signal is





largest in boreal summer due to adjustments in the northern extratropics for both posterior fluxes (**Figures 14** and **S15**). As seen in **Figure 7**, these adjustments in July are much greater when GOSAT data is assimilated. Indeed Byrne et al. (2017) found large sensitivity of boreal summer fluxes to GOSAT data. This is also consistent with the large summertime flux adjustments of Liu et al. (2014) and the increased European fluxes seen from May-August in Houweling et al. (2015).

**Figure 15** compares the regional contributions to the global flux signals for the two models. The differences seen in the UTLS in **Figure 13c** are evidently due to differences seen in the northern extratropics (**Figure 15c**) in boreal summer and autumn. Since GEM-MACH-GHG agrees better with HIPPO-3 in the middle and upper troposphere and in the lower stratosphere, it is possible that its signal is more accurate in this region. However, given the limited temporal and spatial domain of the measurements, such a conclusion would be tentative at best. **Figure 15** also shows that when the global mean

is subdivided into three zonal bands, a tiny (negligible) influence of atmospheric transport errors associated with imperfect meteorology becomes apparent near the surface (**Figure 15a-b**) in the northern extratropics during boreal spring and summer. In addition, the $CO_2$ uncertainty due to wind field uncertainty exceeds the flux signal obtained from assimilating either set of observations in the tropical upper stratosphere (not shown). Overall, however, the global flux signals are very similar between the two models, even after dividing them into regional contributions.

**3.2.5 Zonal asymmetry in the flux signals**

Departures from zonal mean flux signals can be used to examine shorter temporal and spatial scales in the flux signal. The zonal mean flow has no zonal standard deviation (by definition) so large zonal standard deviations indicate greater zonal structure (or asymmetry within a zonal band). Moreover, once the flux signal has diffused to the background (or zonal mean) state, it will not contribute to the zonal standard deviation. As noted earlier, the flux signal diffuses to the background

state in about 3 months. Thus the zonal standard deviation field shown in **Figure 16** reflects shorter time scales than does the zonal mean of the flux signal. That explains why curves in **Figure 16** do not have a trend in the troposphere as was seen in **Figures 13-15**. The zonal structure is largest in boreal summer in the lower troposphere (black curves in panels a-b) mainly due to the flux signal in the northern extratropics (red curves). This suggests GOSAT is capable of picking up finer spatial scales due to the high density of observations in this region when the satellite shifts its view to the northern

hemisphere (**Figure 2**). The impact of large flux increments in boreal summer was also seen in zonal mean fields in **Figures 7** and **12**. In addition, a rather constant and large zonal standard deviation is seen in the tropics (blue curve in **Figure 16a**). This is consistent with the findings of Deng et al. (2016) and Byrne et al. (2017) that finer scale flux estimates can be obtained in the tropics with GOSAT glint observations. However, in the middle troposphere and above, the seasonal variation in zonal standard deviation diminishes, as occurred with the zonal mean flux signal (**Figures 13-15**). Also the

magnitude of the zonal standard deviation diminishes with height. In the stratosphere, while the magnitudes are small, a small trend is seen in the second year in panels d and e. This suggests that after one year of simulation some zonal



asymmetry is being seen in the flux signal and that transit times of surface flux perturbations to the stratosphere are longer than the three months needed to reach the mid-troposphere. This delayed response makes sense given that the mean age of air is about one year in the tropical lower stratosphere and increases to more than four years in the extratropical lower stratosphere (Andrews et al., 2001; Waugh and Hall, 2002). Thus perturbations of stratospheric flow can be expected to

have a delayed response to perturbations in surface fluxes.

**Figure S16** is comparable to **Figure 16** but for the in situ-based fluxes. As with GOSAT data, seasonal variation in flux signal is also seen in the lower and middle troposphere in the northern extratropics. There is also a seasonal variation in the zonal standard deviations in the tropics (**Figure S16a-b**). Spatial variations in the tropics are larger in boreal summer as well as in March  2011. The March 2011 event was also seen with GOSAT data and with both models (Fig. 18f) and may be

related to the fact that enhanced $CO_2$ in tropical Asia was seen in commercial aircraft based in situ data in March to May 2011 (Basu et al., 2014, their figure 3). As with the GOSAT-based flux signal, the magnitude of zonal standard deviations diminishes with altitude, and in the stratosphere, a trend in values is seen (**Figure S16d-e**). The differences in zonal asymmetry of flux signal seen with the two observing systems are directly compared in **Figure 17**. Now it is clear that more zonal structure is apparent with GOSAT data in the lower and middle troposphere (**Figure 17a-b**). Also, the slightly greater

zonal structure in stratospheric increments obtained with in situ data in the first year is also evident (**Figure 17d**). However, the flux signal in the stratosphere due to the assimilation of observations does not exceed that due to wind field uncertainty in the middle and upper stratosphere (**Figure 17e-f**). In the lower stratosphere (**Figure 17d**), the zonal structure in the first year is also not to be trusted. In the lower troposphere, zonal asymmetry in GOSAT flux signals exceed that arising from wind field uncertainty except in November, December and January (**Figure 17a**). However, for in situ data, the zonal

structure can only be trusted in boreal summer (June, July and August). Thus the satellite data are potentially able to retrieve fluxes on finer spatial scales than are in situ data through most of the year. Given the difference in observation densities (**Figures 1** and **2**), this result is not surprising. The lack of ability of in situ data to produce zonal asymmetry in flux signals that are larger than those arising from uncertainty in wind fields outside of boreal summer may indicate why it has been difficult for flux inversions to regionally attribute sources with this observation network (e.g. Gurney et al., 2002, Peters et

al., 2010, Bruhwiler et al., 2011, Peylin et al., 2013).

Contributions of the 3 zonal bands to the globally averaged zonal standard deviations are shown in **Figure 18**. In the northern extratropics, GOSAT data produce zonal structures that exceeds errors due to wind field uncertainty from May to October in the lower troposphere (**Figure 18a**), from June to September in the middle troposphere (**Figure 18b**) and in July and August in the upper troposphere (**Figure 18c**). However, the in situ data produce zonal structure that cannot be trusted

except in July, August and September in the lower troposphere (**Figure 18a**). In the tropics, zonal structure is evident in $CO_2$ fields forced by GOSAT posterior fluxes in the lower troposphere at all times (**Figure 18d**). In the middle troposphere,




the tropical zonal structure can be trusted in August, September, October (**Figure 18e**). For the $CO_2$ fields informed by in situ observations, the zonal structure in the tropics is trustable only in July, August and September in the lower and middle troposphere (**Figure 18d-e**). In August and September 2010, in the upper troposphere (**Figure 18f**), both GOSAT and in situ data produce zonal structure that exceeds that arising from uncertain wind fields. Both models also produce qualitatively

similar results with the exception of the tropical lower troposphere (**Figure 18d**) and the UTLS region in the second year (**Figure 18c, i**) where GEM-MACH-GHG produces more zonal structure. Given the much higher resolution (horizontally and vertically) of this model, it can generate finer scale structures from the coarse resolution fluxes that eventually propagate to the stratosphere. The differences may also be due to the higher resolution of GEM-MACH-GHG directly producing spatial variations in UTLS flow and in the tropics.

In this subsection, the zonal standard deviations of the flux signal were examined in a global sense and in terms of contributions to the global values. The potential benefit of the higher density GOSAT observations is clearly evident in enhanced zonal structures particularly in the northern extratropics in boreal summer and in the tropics, year round. These values exceed the uncertainty in $CO_2$ due to uncertain meteorology much of the time. However, these diagnostics can only indicate a potential benefit since the increased zonal variation was not validated against independent measurements. While

this type of validation is not yet possible because it requires high resolution, globally distributed, independent measurement networks, Houweling et al. (2015) found that flux inversions with GOSAT data do not agree with each other on subcontinental scales. They conclude that flux inversions using GOSAT data do not sufficiently constrain regional scale fluxes.

**4 Summary and Discussion**

In this work, we have examined how fluxes retrieved with the GEOS-Chem flux inversion system and in situ and GOSAT observations inform $CO_2$ model simulations throughout the troposphere and lower stratosphere. We defined the flux signal as the $CO_2$ distribution obtained with posterior fluxes minus that obtained with prior fluxes. The flux signal therefore reflects the influence of observations in the recent past as well as the accumulated influence of observations in the distant past. By definition, the flux signal is a function of both the transport model and the prior fluxes used. The largest

contribution to the global flux signal in the troposphere is from the northern extratropics but the stratospheric signal primarily reflects tropical influence (**Figure 12**). The global flux signal due to GOSAT observations has much stronger seasonal variations than that due to in situ observations (**Figure 13**). Furthermore, a difference of about 0.5 ppm is seen between the simulations obtained using GOSAT and in situ posterior fluxes with the latter agreeing better with observations (TCCON, HIPPO-3 in the northern extratropics above the middle troposphere, and NOAA aircraft on annual time scales)

(**Figures 8, 10, S10, S12, S14**). The inversion constrained by GOSAT data does not recover the global mean flux as well as



the in situ inversion. However, GOSAT-informed $CO_2$ distributions can be revealed to better capture the seasonal cycle at most northern extratropical TCCON sites (**Figure 9, S12**). Zonal standard deviations of the flux signal (which reveal spatial structures in the zonal direction) are much larger when GOSAT-informed posteriors are used (in the northern extratropics outside of boreal winter and in the tropics throughout the year) (**Figure 16, 17**). This indicates a potential for GOSAT data

to retrieve finer scale fluxes since the accuracy of such finer scale features requires a dense network of independent measurements to validate.

Since the flux signal depends on the transport model used, we used two different models (GEOS-Chem and GEM-MACH-GHG) to define the flux signal. Since GEOS-Chem was used for the flux inversions, subsequent integrations of posterior fluxes are consistent with the transport assumed during the flux inversion. However, the posterior $CO_2$ distributions

obtained with GEM-MACH-GHG convolve its transport model error with that of GEOS-Chem. Indeed, a difference in model transport times to the southern hemisphere was seen. Yet despite this caveat, all of the main conclusions held for both models. Moreover, the use of GEM-MACH-GHG, which is a coupled meteorology-tracer transport model, permitted the calculation of uncertainties in posterior $CO_2$ distributions due to uncertain wind fields. Actual meteorological analysis errors were used to perturb wind fields and repeat all simulations (see supplemental material). The impact of perturbed wind fields

on $CO_2$ distributions was used to define a minimum level of uncertainty (since in reality, model integrations of $CO_2$ will also include errors from fluxes, model formulation and representativeness as well as the inevitable imperfections from meteorological analyses). This error was useful for determining when spatial scales (departure from zonal symmetry) could be trusted although, being a minimum error, it provides an optimistic assessment. In situ observations were found to generate zonal standard deviations larger than this minimum level only in boreal summer whereas GOSAT data exceeded

this threshold through most of the year (**Figure 16, 17**). This potential for retrieving finer spatial scales with GOSAT sampling relative to the in situ network makes sense given the density of GOSAT observations (**Figure 2**) and is consistent with the prediction of Takagi et al. (2014) or Deng et al. (2016). Moreover, the ability to retrieve zonal structure is evident throughout the year in the tropics and in all seasons except boreal winter in the northern extratropics is rather encouraging. However, verifying such finer scales will be challenging given the limited spatial coverage of validating measurements from

TCCON or aircraft platforms and temporal and spatial scales resolved may depend on the characteristics of the flux inversion system. Indeed, the current dispute over the enhanced European sinks obtained with GOSAT data (Feng et al., 2016; Reuter et al., 2014; Houweling et al., 2015) indicates that the finer spatial scales retrieved are not necessarily correct and are difficult to validate. Furthermore, the fact that the spatial structure seen in flux signals obtained with in situ data surpassed the minimum uncertainty level only in boreal summer implies that regional attribution of fluxes may be

challenging with the in situ observation network alone when the inversion integrates signals over many seasons. Because our uncertainty arises from imperfect meteorological analyses, its impact cannot be seen in flux inversions obtained from a



single model forced by a single set of driving meteorological fields. However, this error source should be evident in multi-system comparison studies when the systems use different sources of meteorological fields.

By examining the behaviour of each observing system separately, it was possible to isolate differences in their impact on posterior fluxes obtained with our flux inversion system. In particular, it is found that the in situ observing system results in

posterior fluxes that well define the global mean $CO_2$ on annual time scales and that there is a dependence of seasonal variations of the global flux signal on observation system. However both systems defined the annual budget for 2010 equally well. GOSAT was also shown to potentially better retrieve finer spatial scales within a zonal band. The importance of these results is two-fold. First, the implications are that caution should be exercised when drawing conclusions based on sub-annual variations of the global mean $CO_2$ because they depend on the observation sets used. Since $CO_2$ has strong

seasonal variations, the flux signal in the lower atmosphere should also have seasonal variations if the prior fluxes have errors on seasonal timescales (e.g. as in Liu et al., 2014, or Ott et al., 2015). The challenge is that the seasonal variation of GOSAT data coverage will be convolved with an actual seasonal variation of fluxes. Second, our results identify spatial scales of atmospheric $CO_2$ that are best constrained by each observing network, in the context of our flux inversion system. Specifically, the in situ network captures global mean (and the 18-month mean at most TCCON stations) well, while

GOSAT better captures zonally asymmetric structures and the seasonal cycle at northern extratropical TCCON sites. Understanding the time scales resolved by different observing systems will be critical for the $CO_2$ assimilation problem with coupled meteorological and GHG transport models at operational centers which are geared toward short assimilation windows (e.g. Polavarapu et al., 2016; Agusti-Panareda et al., 2014; Massart et al., 2016; Ott et al., 2015). For such systems, long time scale information will be challenging to extract from observations and may require novel multi-time scale analysis

approaches.

While our results regarding the behaviour of each observing system has important implications for flux estimation, they must be seen in the context of the inversion system used, namely, GEOS-Chem and 4D-Var with long assimilation windows. Aspects of the inversion system may impact the results. For this reason, repetition of our experiments with other inversion systems is desirable to determine the generality of results across inversion systems. Furthermore, we suggest that comparing

flux signals obtained by integrating a single model with known transport behaviour with posterior fluxes from various different inversion systems could be a useful diagnostic because it will identify relative mismatches of transport times between models. For example, CT2013B fluxes with our weather model (GEM-MACH-GHG) identified a mismatch in transport of midlatitude fluxes in boreal summer to the high Arctic in autumn with TM5 (Polavarapu et al., 2016) as well as a too fast transport of GEOS-Chem from the tropics to the southern hemisphere relative to GEM-MACH-GHG. While this

diagnostic cannot determine which model's transport is correct, if the reference model's transport issues were known (from age-of-air diagnostics, for example), the flux signal comparison offers a fast, simple way to infer transport issues of other





models. However, only obvious transport mismatches would be identifiable. Regional, or shorter timescale transport mismatches would be hard to identify with a sparse verifying observation network. Indeed, as a result of this work, we plan to identify GEM-MACH-GHG's transport issues through age-of-air diagnostics in the future.

Although only GOSAT-based flux inversions were considered here, it is natural to wonder if the results would apply to
OCO-2. Byrne et al. (2017) note that OCO-2 has higher spatial resolution and higher precision (due to aggregation of measurements in 2x2.5 grid) and that OCO-2 is better at picking up NH extratropical fluxes than GOSAT (their Fig. 10). OCO-2 also had the best constraints on regional fluxes in the tropics. It is easy to speculate that even finer spatial scales than seen here with GOSAT data could be expected. However, OCO-2 also has a seasonal variation in coverage which has been shown to produce a bias in global annual flux (Liu et al., 2014). Although Liu et al., (2014), and Houweling et al., (2015)
suggest that flux inversion systems are partly to blame by not permitting seasonal correlations of covariances, it may be desirable to obtain additional measurements of the northern hemisphere during boreal winter. GOSAT, OCO-2 and TanSat measure in the shortwave infrared range so their latitudinal coverage does vary seasonally. The seasonal variation of coverage could be reduced if more nadir observations over snow covered regions were processed for the winter or more ocean glint observations were made in winter. (However, signal-to-noise ratio for the $CO_2$ bands is lower over snow, so
retrieving over snow will typically result in poorer precision than over other surfaces.) Furthermore, active measurements such as Active Sensing of CO2 Emissions over Nights, Days and Seasons (ASCENDS) (https://decadal.gsfc.nasa.gov/ascends.html) that do not depend on sunlight would complement the current network of in situ and satellite measurements.

In this work, we have separately considered the impact of in situ and GOSAT data on posterior $CO_2$ distributions in order to
better understand the behaviour of each type of observation in the context of a flux inversion and modelling system. Ultimately, the best network will be a combination of both types of observation (Baker et al., 2006). By revealing the complementary benefits of the two types of observations, our results indicate a need for further research to understand how best to adapt flux inversion systems to take advantage of each type of observation. For example, in situ data could constrain biases in satellite data as in Feng et al. (2016) but perhaps also the long time scale global mean, with satellite data being used
to improve regional scale fluxes.

## 5 Appendix A

The GEOS-Chem adjoint model (Henze et al., 2007) calculates the derivative of the modeled $CO_2$ concentration with respect to a set of model parameters, $f$. We use the adjoint model to calculate the sensitivity of modelled $CO_2$ concentrations to an earlier atmospheric $CO_2$ state over a volume of atmosphere with units of parts per million by volume (ppm) and use the





adjoint model to calculate the gradient $\nabla_f J$. For this study, $J$ is defined as the mean $CO_2$ concentration over 20°S-20°N and 500-250 hPa at instantaneous time $t_0$:

$$J = \left[ \sum_{k=500hPa}^{250hPa} \sum_{j=-20°}^{20°} \sum_{i=0°}^{360°} \frac{C_{i,j,k,t_0}}{M_{i,j,k,t_0}} \right] \cdot 10^6 \qquad \text{(A1)}$$

where $C_{i,j,k,t_0}$ and $M_{i,j,k,t_0}$ are the molar abundances of $CO_2$ and air at longitude i, latitude j, level $k$, and time $t_0$. Gas

abundances are obtained by sampling a forward model simulation at the time $t_0$. The sensitivity is obtained by calculating the gradient of $J$ with respect to an earlier atmospheric $CO_2$ state, $f_{i,j,k,t}$:

$$\gamma_{i,j,k,t} = \frac{\partial J}{\partial f_{i,j,k,t}}. \qquad \text{(A2)}$$

**Acknowledgements**

Work at the University of Toronto was supported by funding from the Canadian Space Agency, Environment and Climate
Change Canada, and the Natural Science and Engineering Research Council of Canada. We are grateful to Ray Nassar and Douglas Chan for helpful comments on an earlier version of this manuscript. ACOS GOSAT data were produced by the ACOS/OCO-2 project at the Jet Propulsion Laboratory (JPL), California Institute of Technology, and obtained from the JPL website, co2.jpl.nasa.gov. We acknowledge the GOSAT Project for acquiring these spectra. TCCON data were obtained from the TCCON Data Archive, hosted by the Carbon Dioxide Information Analysis Center (CDIAC) at
http://tccon.ornl.gov/. We thank TCCON PIs Paul Wennberg, Caltech (Lamont, Park Falls), David Griffith, University of Wollongong (Darwin and Wollongong), Justus Notholt, University of Bremen (Bremen), Nicholas Deutscher, University of Bremen (Bialystok), Thorsten Warneke, University of Bremen (Orleans), Dave Pollard, NIWA (Lauder), Ralf Sussmann, IMKIFU (Garmisch), Kimberly Strong, University of Toronto (Eureka), Rigel Kivi, FMI (Sodankylä), Frank Hase, KIT (Karlsruhe), and Matthias Schneider, KIT (Izaña). We are grateful to Colm Sweeney (NOAA ESRL) for providing the
NOAA aircraft profiles and to Ken Masarie of the NOAA Global Monitoring Division in Boulder, Colorado for compiling ObsPack2013. The National Oceanic and Atmospheric Administration (NOAA) North American Carbon Program has funded NOAA/ESRL Global Greenhouse Gas Reference Network Aircraft program. The ObsPack data were obtained for the period 2000-2012 (obspack_co2_1_PROTOTYPE_v1.0.4_2013-11-25) from http://dx.doi.org/10.3334/OBSPACK/1001. We are grateful to NSF and NOAA for producing and providing HIPPO-2 (http://hippo.ucar.edu) aircraft measurements
(HIPPO Merged 10-second Meteorology, Atmospheric Chemistry, Aerosol Data (R_20121129)). We would like to thank




Doug Worthy of Atmospheric Science and Technology Directorate (ASTD), Environment and Climate Change Canada, for developing and maintaining ECCC's greenhouse gas measurement network and for providing the $CO_2$ concentration measurement data.

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




| Reference pressure (hPa) | GEOS-Chem Model level ref | | GEM Model level ref | |
|---|---|---|---|---|
| | index | hPa | index | hPa |
| 850 | 9 | 856.781 | 69 | 854.893 |
| 500 | 22 | 503.795 | 57 | 501.327 |
| 250 | 28 | 263.587 | 47 | 258.932 |
| 100 | 34 | 99.191 | 34 | 99.1268 |
| 33 | 38 | 33.814 | 19 | 32.9691 |
| 7 | 41 | 6.588 | 10 | 6.86514 |

Table 1: Comparable model levels used in later figures. An approximate pressure level is computed for each model level assuming a reference surface pressure of 1000 hPa.





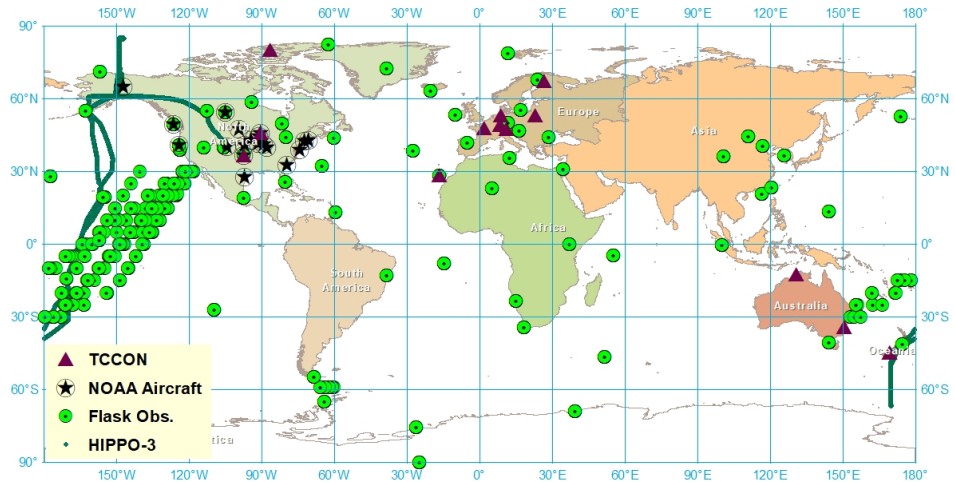

**Figure 1.** In situ observation network and observations used for verification. The in situ observations used in the GEOS-Chem flux inversion are indicated in green circles. Observations used for model assessment are also shown: TCCON (triangles), NOAA aircraft (stars) and HIPPO-3 aircraft (green line).



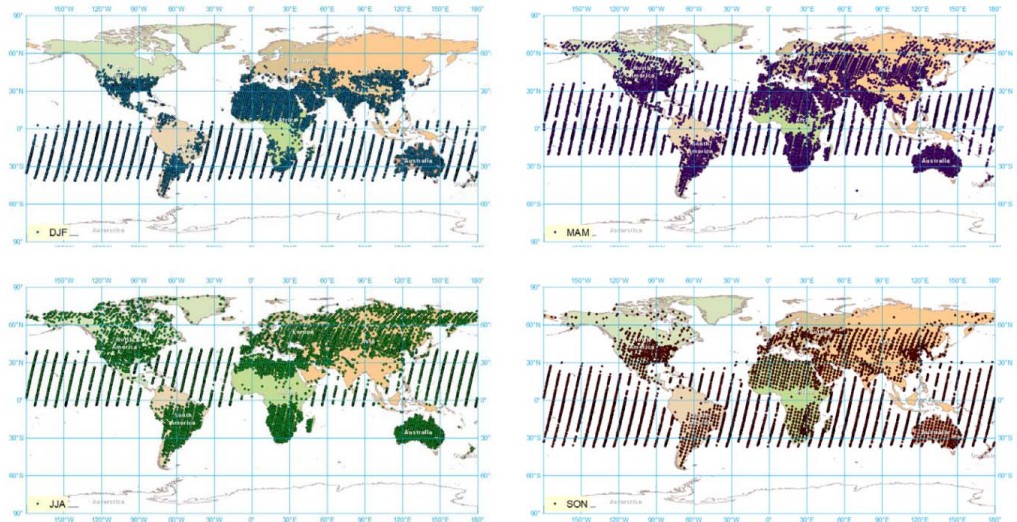

**Figure 2.** Seasonal variation of GOSAT observations. The observations used in the GOSAT-based flux inversions are shown for four seasons: boreal winter (December, January, February – top left), boreal spring (March, April, May – top right), boreal summer (June, July, August – bottom left) and boreal autumn (September, October, November – bottom right) for 2010.





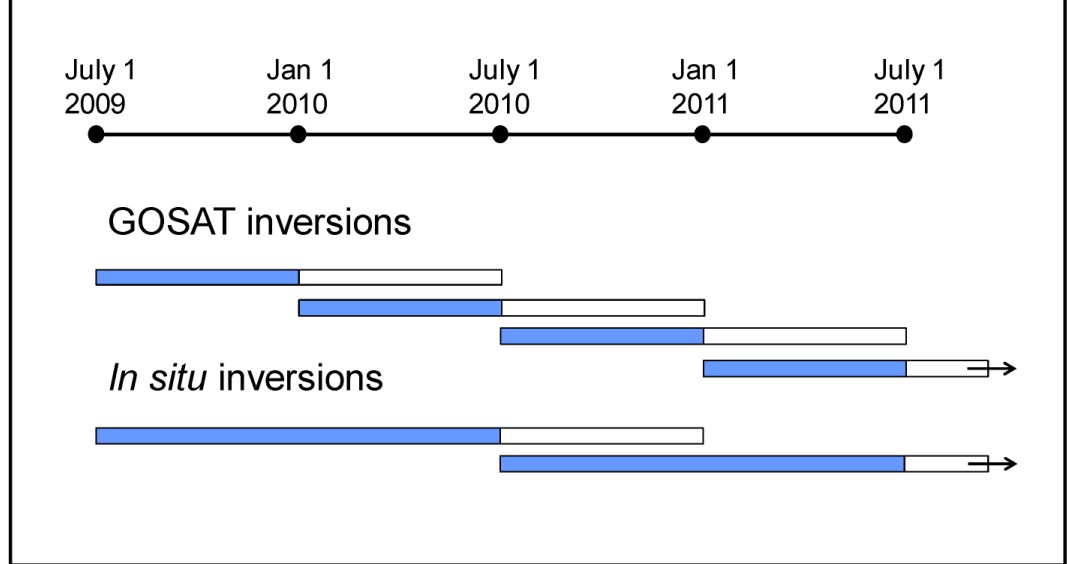

**Figure 3.** Schematic diagram of GEOS-Chem inversion experiments. The inversions involving the assimilation of GOSAT data were done in four 12-month segments. The fluxes obtained from the first 6 months of each segment were retained as the retrieved fluxes. The inversions involving in situ data were done in two 18-month segments with the fluxes retained from the first 12 months. Thus retrieved fluxes were available for the 24 months from 1 July 2009 to 30 June 2011 for both sets of flux inversions.



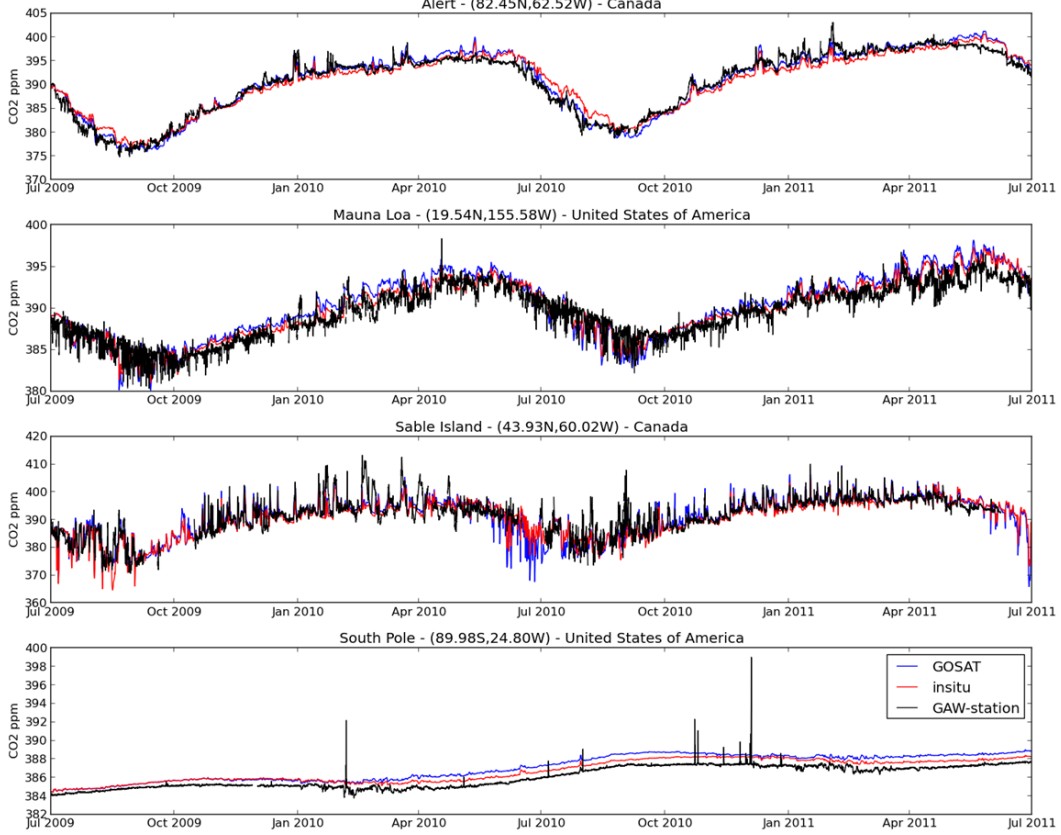

**Figure 4.** Time series of $CO_2$ observations and GEM-MACH-GHG model simulations for 1 July 2009 to 1 July 2011. The $CO_2$ observations are from ECCC GHG in situ measurement network for Alert (top), Mauna Loa (second), Sable Island (third) and South Pole (bottom). The observations are indicated in black. The model simulations used posterior fluxes obtained from inversions with GEOS-Chem using in situ (red curves) or GOSAT (blue curves) observations.





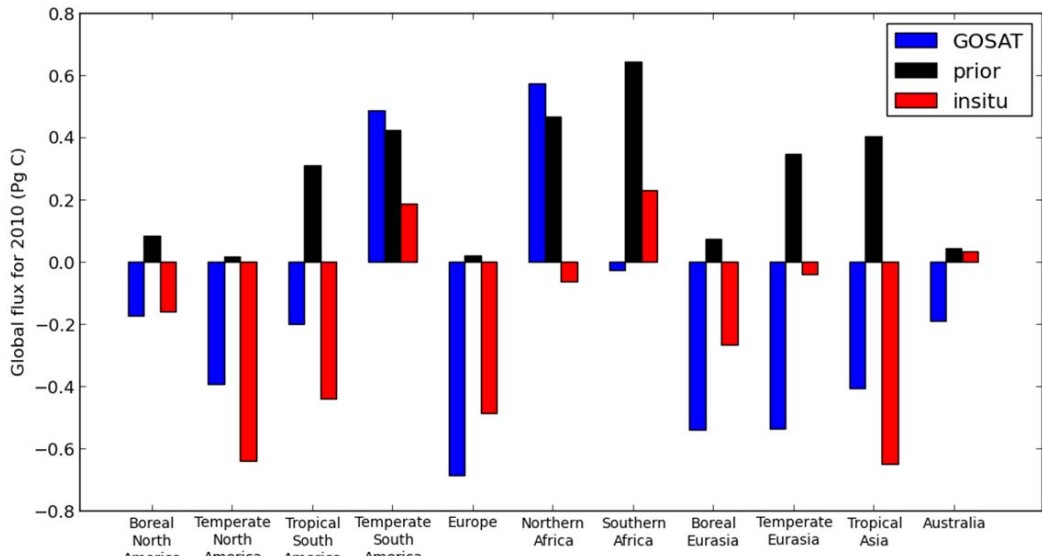

**Figure 5.** Annual fluxes for 2010 for the 11 TransCom regions inferred by GOSAT (blue) and in situ (red) data. Also shown is the prior flux (black). Values include natural fluxes as well as biofuel and biomass burning emissions.





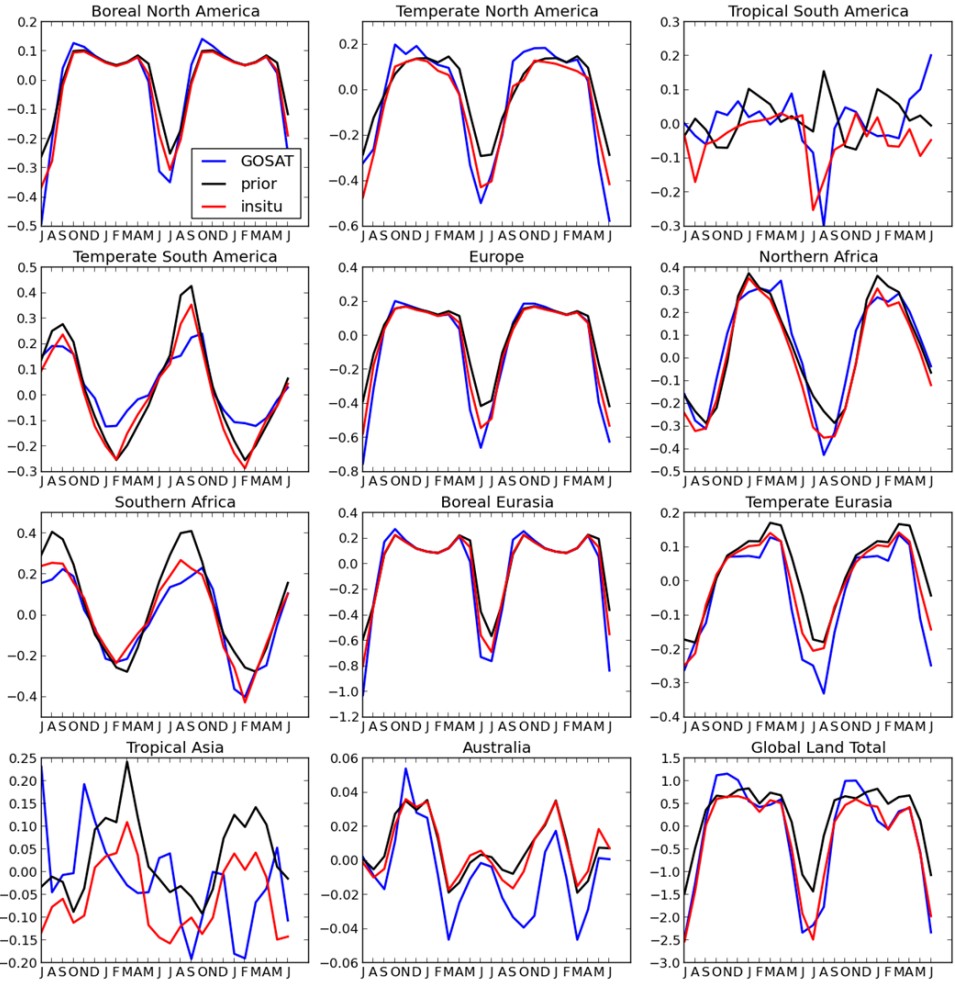

**Figure 6**: Monthly fluxes in PgC yr$^{-1}$ for the 11 TransCom land regions and the global land total retrieved from GEOS-Chem 4D-Var inversions using GOSAT (Blue) or in situ (red) data. Also shown are prior fluxes (black). The full simulation period of July 2009 to June 2011 is shown, including the spin up period (July – Dec. 2009).





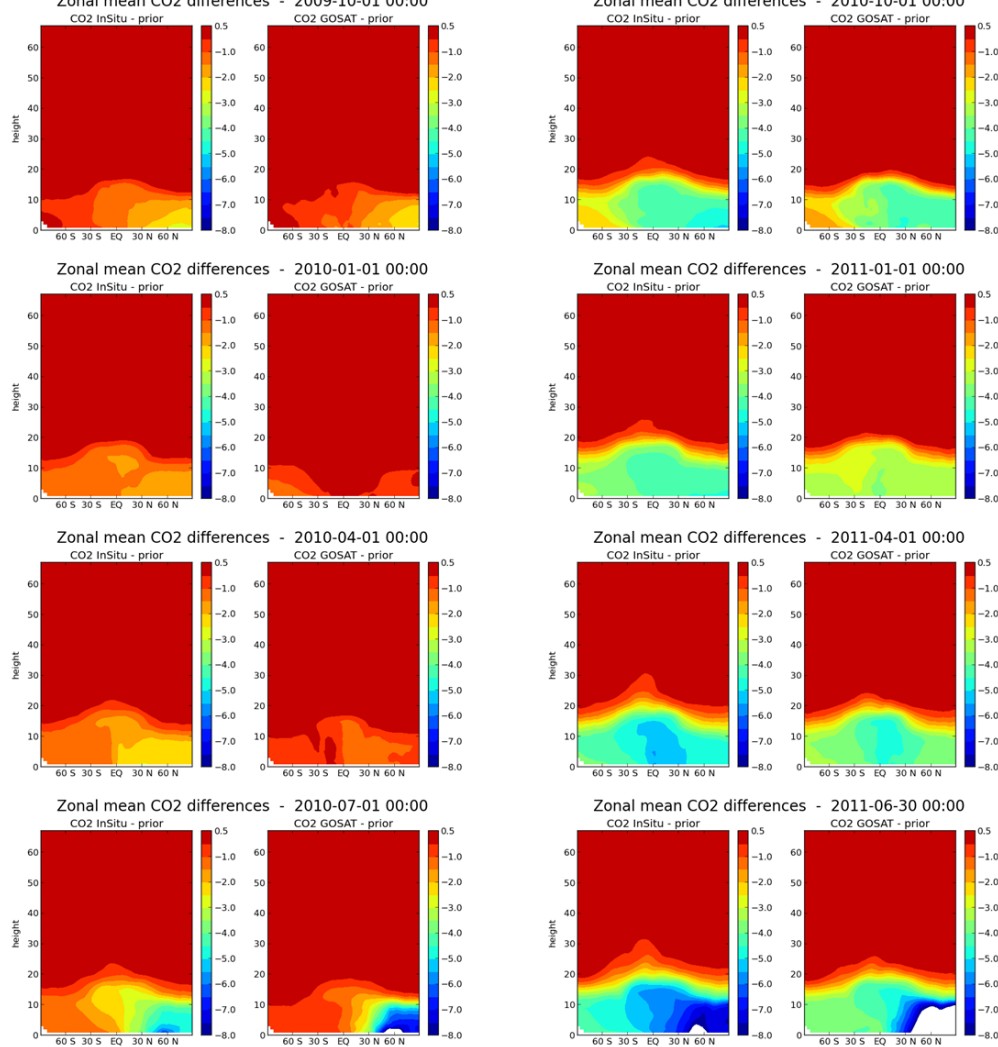

**Figure 7.** Time sequence of zonal mean flux signals simulated with GEOS-Chem. Zonal mean fields are displayed as a function of height and latitude in units of ppm. Shown are the in situ (leftmost of each pair) and the GOSAT (rightmost of each pair) zonal mean flux signals. The earliest date is in the top left corner with subsequent dates following down the left side then continuing down the right side. Dates are indicated above each pair of panels starting on 1 October 2009 and continuing in three-month intervals to 30 June 2011.







**Figure 8.** Comparison of GEOS-Chem $CO_2$ simulations with GOSAT-derived (black) and in situ (red) derived posterior

fluxes to TCCON measurements at 14 sites (Darwin, Wollongong, 2 instruments at Lauder, Izaña, Lamont, Park Falls,

5   Garmisch, Orléans, Karlsruhe, Bremen, Bialystock, Sodankylä, and Eureka).   Stations are ordered by latitude from

southernmost to northernmost.  The mean residual in ppm was computed for each stations from December 2009 to May

2011, inclusive.  Positive values mean the modelled $CO_2$ is generally higher than observed $CO_2$.





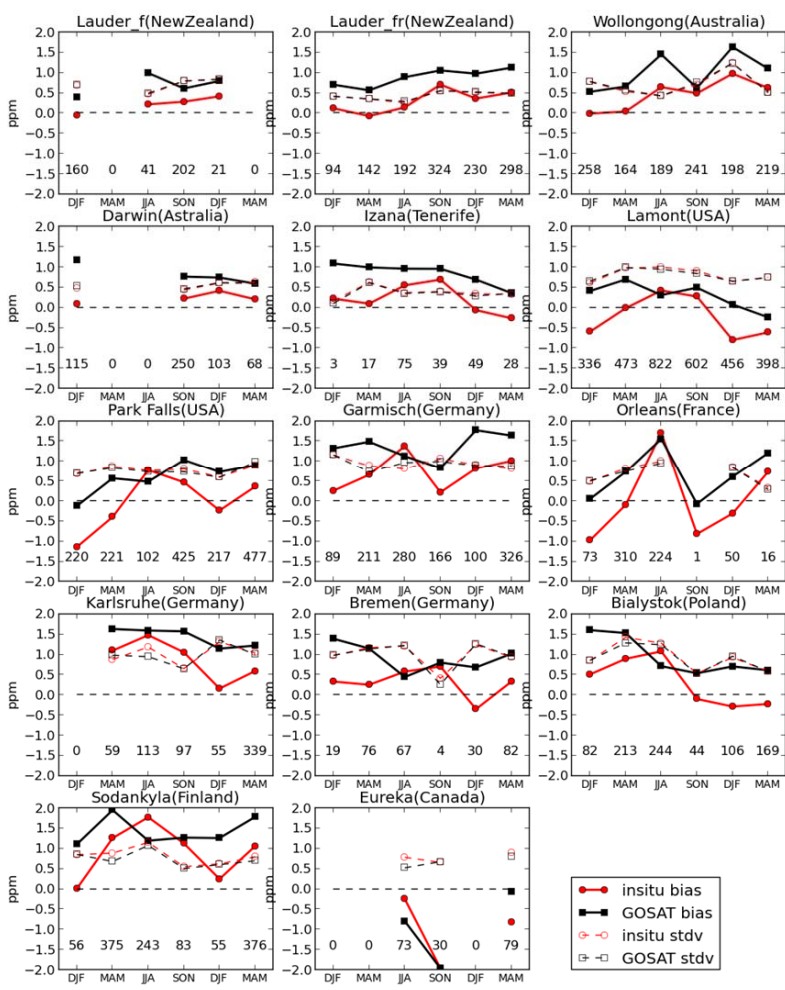

**Figure 9.** Comparison between TCCON measurements at 14 sites and the GEOS-Chem $CO_2$ simulations driven with posterior fluxes from the GOSAT (black) and in situ (red) inversions. Scores (bias and standard deviation) are aggregated by three-month seasons from December 2009 to May 2011. Lauder appears twice because there are two different instruments there.



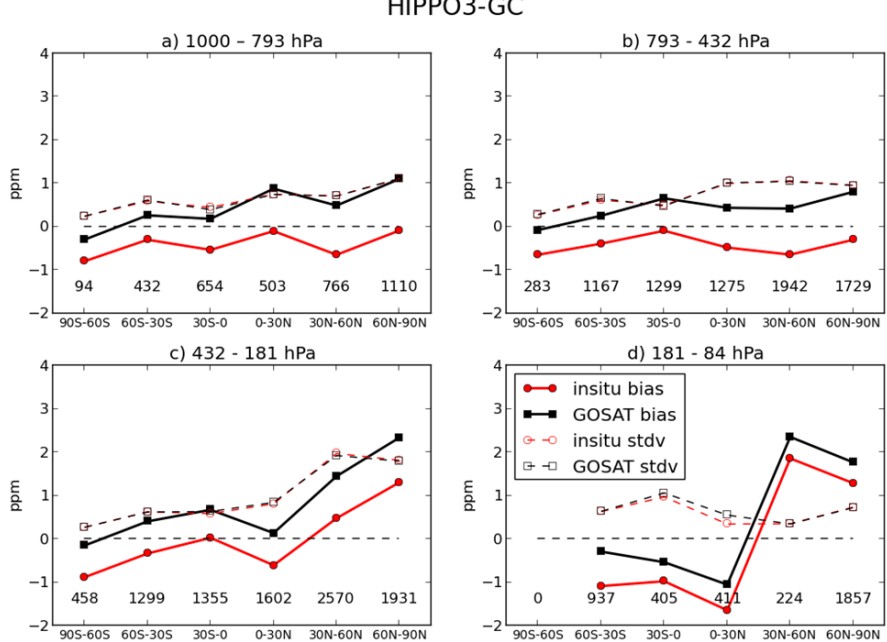

**Figure 10.** Comparison between the HIPPO-3 measurements and the GEOS-Chem $CO_2$ simulations driven with posterior fluxes from the GOSAT (black) and in situ (red) inversions. Scores (bias and standard deviation) of modelled minus observed values are aggregated by latitude band and over the pressure layers given above each panel. The numbers of observations used in each statistic are indicated within each panel. The flights occurred between 24 March to 16 April 2010.





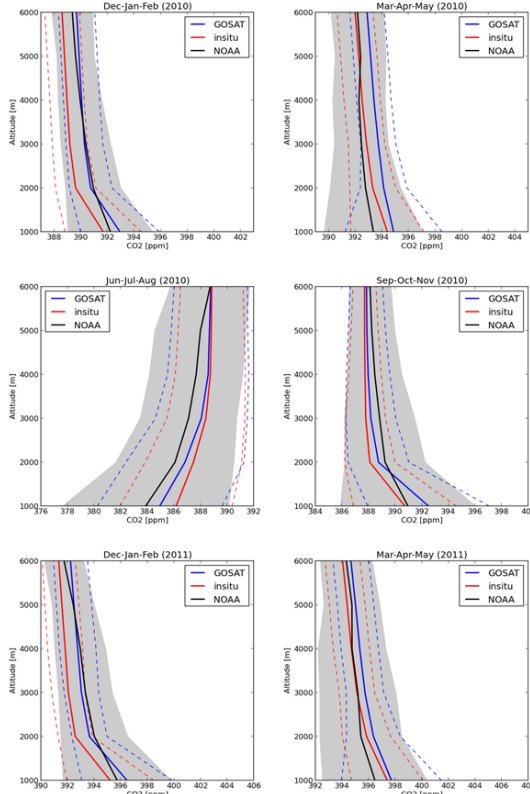

**Figure 11.** Comparison of mean GEOS-Chem model profiles of $CO_2$ to NOAA aircraft observations. Observations (black curves) are from obspack_co2_1_PROTOTYPE_v1.0.4_2013-11-25 for locations over continental U.S. and Canada, only. Observed and modelled profiles are binned over 3-month seasons as indicated above each panel. Model simulations used

5   posterior fluxes from GEOS-Chem inversions with GOSAT (blue) or in situ (red) observations. The shaded grey regions indicate plus or minus one standard deviation for the observations while the dashed coloured lines indicate the same quantities but for the different model runs. Sites used are: Beaver Crossing, Nebraska; Bradgate, Iowa; Briggsdale, Colorado; Cape May, New Jersey; Charleston, South Carolina; Dahlen, North Dakota; East Trout Lake, Saskatchewan; Estevan Point, British Columbia; Fairchild, Wisconsin; Harvard Forest, Massachusetts; Homer, Illinois; Oglesby, Illinois;

10  Park Falls, Wisconsin; Poker Flat, Alaska; Sinton, Texas; Southern Great Plains, Oklahoma; Trinidad Head, California; West Branch, Iowa; Worcester, Massachusetts.





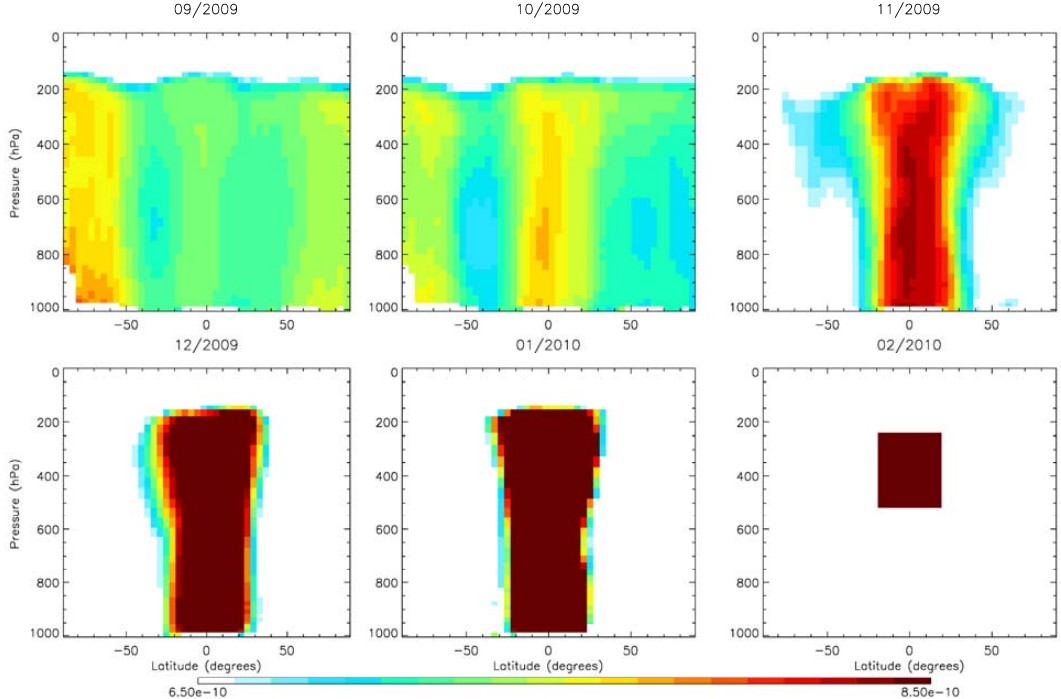

**Figure 12**: Sensitivity (in ppm/ppm) of the GEOS-Chem tropical tropospheric $CO_2$ on 1 February 2010 to the 3D modeled
state on earlier dates. Sensitivity fields are zonally averaged instantaneous fields for the first day of each month from
September 2009 to February 2010 in the various panels.





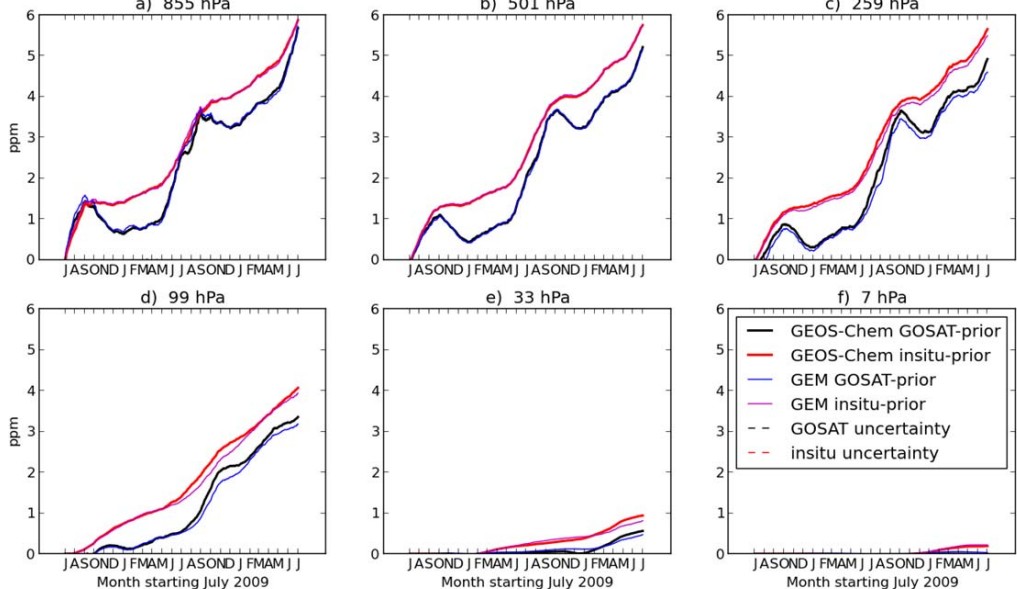

**Figure 13.** Global mean $CO_2$ flux signal for 1 July 2009 to 30 June 2011 from the GOSAT-based posterior fluxes (solid black curves) and the in situ-based posterior fluxes (solid red curves). Flux signals (prior minus posterior $CO_2$ fields) are shown for the model level closest to the nominal pressure level indicated above each panel for both GEOS-Chem (thick lines) and GEM-MACH-GHG (thin lines). The global mean of the $CO_2$ uncertainty is shown for the GOSAT posterior flux integration (black dashed curves) and the in situ posterior flux integration (red dashed curves). Uncertainty in $CO_2$ is estimated with GEM-MACH-GHG by perturbing the meteorological analyses and computing the difference from the unperturbed integration.



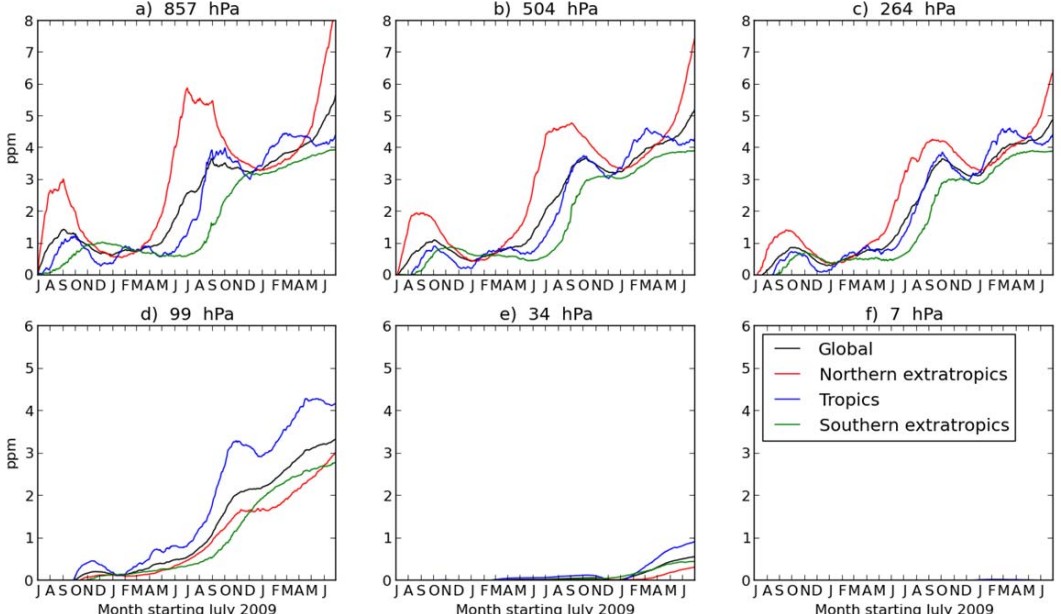

**Figure 14.** Global mean $CO_2$ flux signal obtained with GEOS-Chem with GOSAT-based posterior fluxes. Flux signals (prior minus posterior $CO_2$ fields) are shown for the model level closest to the nominal pressure level indicated above each panel. The coloured curves represent the global total (black) and the contributions to this from the various subregions: northern extratropics (red), southern extratropics (green) and tropics (blue). Because the subregions were chosen to have equal areas, the contribution depicted for each subregion was scaled by a factor of three so that the mean of the contributions from the subregions gives the total contribution.



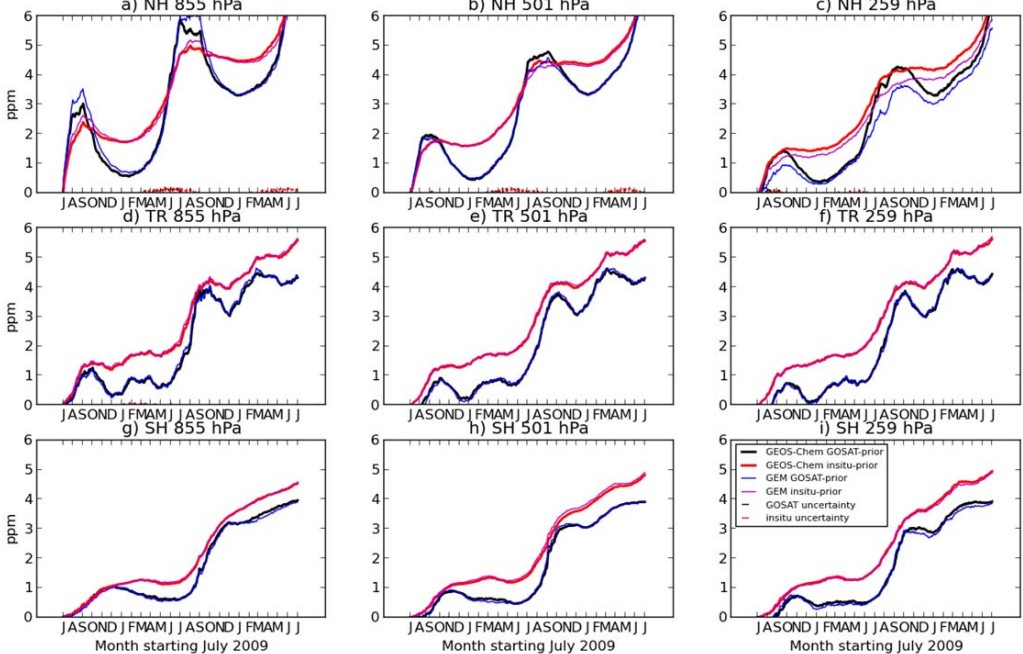

**Figure 15.** Regional contributions to the global mean $CO_2$ flux signal for 1 July 2009 to 30 June 2011. The flux signal is from the GOSAT-based posterior fluxes (solid black curves) and the insitu-based posterior fluxes (solid red curves). Flux signals (prior minus posterior $CO_2$ fields) are shown for the model level closest to the nominal pressure level indicated above each panel for both GEOS-Chem (thick lines) and GEM-MACH-GHG (thin lines). Regional contributions have been multiplied by a factor of three as in Figure 14. Uncertainty in global mean CO2 is shown for the GOSAT posterior flux integration (black dashed curves) and the insitu posterior flux integration (red dashed curves). Uncertainty in $CO_2$ is estimated for each integration by perturbing the meteorological analyses and computing the difference from the unperturbed integration.





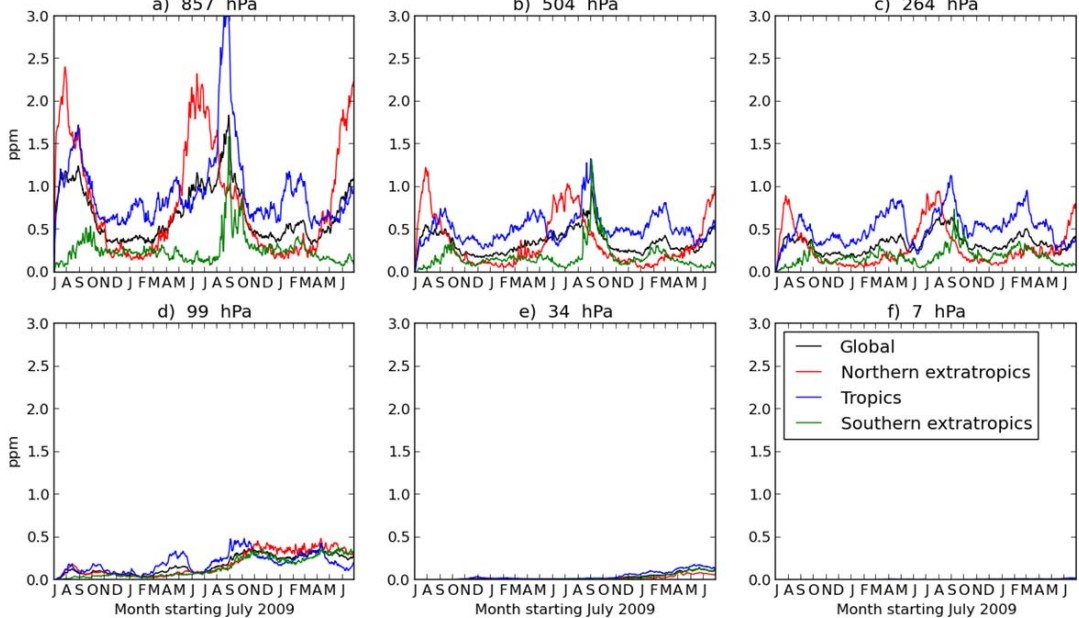

**Figure 16.** Global mean of zonal standard deviation of the $CO_2$ flux signal obtained with GEOS-Chem using GOSAT-based posterior fluxes. Statistics are shown for the model level closest to the nominal pressure level indicated above each panel. The coloured curves represent the global total (black) and the contributions to this from the various subregions: northern extratropics (red), southern extratropics (green) and tropics (blue). Because the subregions were chosen to have equal areas, the contribution depicted for each subregion was scaled by a factor of three so that the mean of the contributions from the subregions gives the total contribution.





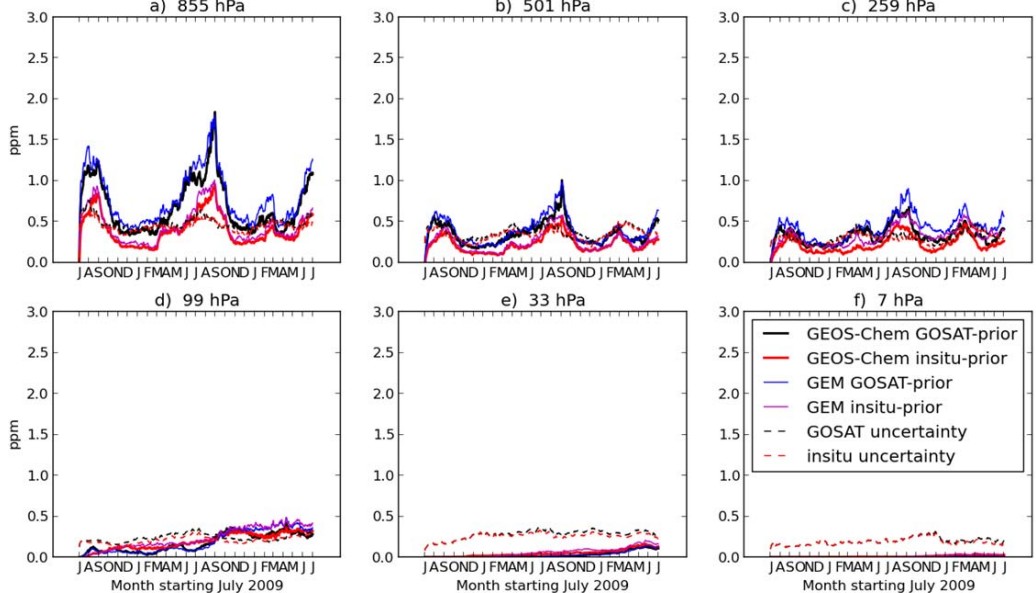

**Figure 17.** Global mean of the zonal standard deviation of the $CO_2$ flux signal for 1 July 2009 to 30 June 2011 from the GOSAT-based posterior fluxes (solid black curves) and the insitu-based posterior fluxes (solid red curves). Flux signals are shown for the model level closest to the nominal pressure level indicated above each panel for both GEOS-Chem (thick lines) and GEM-MACH-GHG (thin lines). The zonal standard deviation of the $CO_2$ uncertainty is shown for the GOSAT posterior flux integration (black dashed curves) and the insitu posterior flux integration (red dashed curves). Uncertainty in $CO_2$ is estimated with GEM-MACH-GHG by perturbing the meteorological analyses and computing the difference in $CO_2$ from the unperturbed integration with a given set of posterior fluxes.




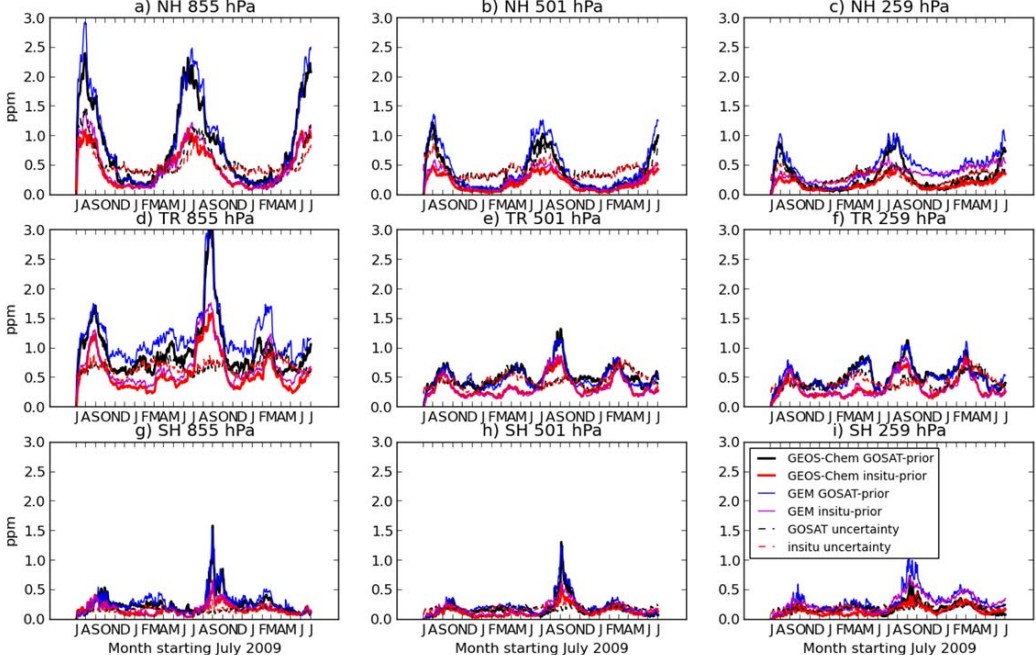

**Figure 18.** Regional contributions to the global mean of the zonal standard deviation of the $CO_2$ flux signal for 1 July 2009

5 to 30 June 2011 from the GOSAT-based posterior fluxes (solid black curves) and the insitu-based posterior fluxes (solid red curves). Flux signals are shown for the model level closest to the nominal pressure level indicated above each panel for both GEOS-Chem (thick lines) and GEM-MACH-GHG (thin lines). Regional contributions have been multiplied by a factor of three as in Figure 12. Uncertainty in zonal standard deviation of $CO_2$ is shown for the GOSAT posterior flux integration (dashed cyan curves) and the insitu posterior flux integration (dashed magenta curves). Uncertainty in $CO_2$ is estimated for

10 each integration by perturbing the meteorological analyses and computing the difference from the unperturbed integration.