# Peer review of "A comparison of posterior atmospheric CO2 adjustments obtained from in situ and GOSAT constrained flux inversions"

_Atmospheric Chemistry and Physics, 2017_

## Referee Comment (RC1) · Anonymous Referee #1 · 15 Mar 2018

This paper compares the atmospheric distributions of CO2 resulting from two sets of optimized fluxes derived from GEOS-Chem using different observing systems based on in situ data and GOSAT data respectively. The results show the differences in the optimized fluxes and how their correction is transported in the atmosphere. An evaluation of the seasonal cycle and inter-hemispheric gradient is also provided. Finally, the zonal variability of the flux correction signal at different vertical levels (boundary layer, free troposphere and stratosphere) is also explored. The differences between the two sets of posterior fluxes and their atmospheric distributions highlight problems

associated with spatial and temporal coverage of obseving systems and their ability to constrain the surface CO2 fluxes at different temporal and spatial scales. Overall, the results point to the conclusion that the in situ observations do a better job at constraining the fluxes at global and annual time scales, leading to smaller biases in their fit with independent observations. While GOSAT data is able to better capture the seasonal cycle at northern extratropical sites. The paper is well written and well structured. However, I have some concerns on the use of atmospheric differences associated with flux correction patterns to draw conclusions on the potential representation of zonally asymmetric patterns by different observing systems. It is not possible to say that GOSAT is (potentially) better at constraining the zonal patterns without substantiating this with an assessement of the errors in zonal variability based on independent observations (e.g. zonal gradients using TCCON or in situ data). The analysis of the seasonal cycle could also be improved by looking at the seasonal amplitude and phase, instead of just providing seasonal biases which is too qualitative in my opinion. The results and conclusions would also be more robust if more than just one year and a half of data was used.

GENERAL COMMENTS

* The use of CO2 flux signal to denote the cumulative impact of the flux corrections/adjustments in the atmosphere is a bit misleading. A flux signal gives the impression that it is associated to a process or phenomenon, while here it just reflects a correction (or analysis increment) which depends on the specific model, prior flux and observation used. I would think that using the term 'posterior atmospheric adjustment' would be a better term to describe the difference between posterior and prior atmospheric distributions of CO2 or alternatively 'flux correction signal'.

* Spatial variability of flux correction signal in the atmosphere does not necessarily translate in better provision of information by observations nor an improvement in spatial/regional patterns. If the observations are very noisy (e.g. GOSAT has larger errors than the in situ observations assimilated in flux inversion systems) or observations are

not homogeneously distributed (e.g. many more data over land than sea as it is the case in northern extratropical regions) then the flux corrections can create artifacts in the zonal variability which increase the zonal variability but are nevertheless not realistic.

\* The paper would benefit from a better quantification of error reduction at different scales based on TCCON and in situ observations which could be presented in tabular format.

\* The fact that the minimal level of uncertainty in the zonal variability associated with imperfect knowledge of winds is around 0.5 ppm and the global zonal variability of flux corrections is of similar magnitude does not make the posterior zonal flux correction pattern is unreliable. The objective of the flux inversion systems is to reduce the uncertainty of the posterior fluxes and the flux corrections on their own do not necessarily reflect the uncertainty reduction. The posterior zonal patterns should be assessed with independent observations and their standard error compared to the minimal level of uncertainty associated with transport.

SPECIFIC COMMENTS

- Page 7, Line 14: Isn't the uncertainty of 22% associated with NEE very low? - Page 7, Line 19: I would not call GOSAT coverage "dense". - Page 8, Lines 2-3: "Note that ... " sentence is not clear. - Page 9: Please provide a quantitative estimate of standard error and bias per month/season for the surface in situ evaluation in order to assess the seasonal cycle quantitatively. When the bias is shown to be smaller, it would help to know by how much - Page 10, Line 2, Page 11, Line 23: Is the flux correction signal in the atmosphere "propagated" or "transported"? - Page 12, Line 16: Why is GOSAT reducing meridional gradient? From Figs 8 and 10 it looks that the meridional gradient from the GOSAT posterior fluxes is worse than that from the in situ data. - Page 14, Lines 22-25: How do you reconcile this with the larger bias of GOSAT versus TCCON in SH? - Page 16, Line 2: How do you explain that GOSAT produces better fit with

observations in middle to upper troposphere in boreal winter? - Page 21, Lines 11-18: The message that GOSAT observations have the potential benefit of improving the zonal structure seems to be contradicted by the results from flux inversions using GOSAT data published in Houwelling et al (2015). Therefore, the conclusion of the potential benefit of GOSAT highlighted in the abstract can be misleading. - Page 23, Line 15: "GOSAT better captures zonally asymmetric structure ..." should be rephrased as this has not been proven in the paper. - Page 23, Lines 24-27: Note that this type of comparison has already been done by Locatelli et al. (2013, ACP) for CH4. - Page 24, Line 10: .. seasonal correlation of "error" covariances. - Figure 4 and Page 9: It would be good to include GEOS-Chem in Fig. 4.

---

## Referee Comment (RC2) · Anonymous Referee #2 · 23 Mar 2018

Overall impression

======================

According to my understanding, this manuscript addresses two major topics. The first is how adjustments to surface fluxes (posterior minus prior) manifest themselves in the atmosphere. This is done by performing inversions for the first two years of GOSAT data using a variational GEOS-Chem system, and propagating the posterior and prior fluxes through a transport model. Along the way, the authors perform some evaluation

of their inverse results, such as comparison to TCCON and HIPPO. The second is how that manifestation varies if a different higher resolution online atmospheric transport model is used. In my opinion, the authors spend too much time on the first topic and not enough on the second, which makes the work not significant enough for a journal like Atmospheric Chemistry and Physics. If this focus were reversed, or the first topic were explored further (explained below), it would make for a much more interesting and scientifically significant paper.

The authors perform inversions of GOSAT and in situ data for two years, and look at the fluxes and resultant atmospheric $CO_2$ fields in the first two years of GOSAT, primarily focusing on 2010. They use a variational inversion technique using the GEOS-Chem transport model. Their conclusions are very similar to previously published literature, such as Houweling et al (2015), Basu et al (2013), Chevallier et al (2014), which they cite. In fact, a very similar (if not identical) set of inversions was already submitted by some of the co-authors to an intercomparison of GOSAT inversions published by Houweling et al (2015). As far as I can tell, there is nothing new or unique about their inversion or analysis compared to the multitude of GOSAT inversions already published for 2010, and this part of the work does not add to the body of existing knowledge about GOSAT retrievals and derived fluxes in and around 2010. GOSAT has been up for eight years now, and retrievals of column $CO_2$ from GOSAT exist for the majority of that period. I do not understand why the authors have limited their study to the first couple of years of GOSAT data. If the authors want to publish a GOSAT inversion study that would be of value to the scientific community, I would recommend performing a longer term study, such as (say) the inter-annual variability of fluxes as seen by GOSAT, or the longer term trends in atmospheric $CO_2$ and $CO_2$ fluxes as seen by GOSAT. The current inversion study, focused on 2010 (with some padding on either side), is of limited interest.

The second thread in their work, however, is more interesting. They perform forward runs with two different models of atmospheric transport driven by the same fluxes and

look at the difference in the "flux signal" in the atmosphere. The non-GEOS-Chem model is the higher resolution GEM-MACH-GHG, a fairly new addition to this community (Polavarapu et al, 2016). Not only did they transport $CO_2$ with GEM-MACH-GHG, they also perturbed the transport with analysis errors from the meteorological assimilation system, thereby simulating the impact of uncertainties in the met fields on $CO_2$ variations. They derive a "baseline" $CO_2$ variation from this error propagation, contending that variations smaller than this detected by an observing system cannot be reliably ascribed to fluxes. This, to my knowledge, is fairly unique in the tracer transport community, and provides a recipe for deriving transport errors in $CO_2$ space. Such errors can be used, e.g., if GEM-MACH-GHG or a derived offline model is used for trace gas inversions. This technique may also be valid for deriving "baseline" transport errors for an offline model if an ensemble is run for the parent model with greenhouse gases (e.g., GEOS5 for GEOS-Chem).

If the authors would like to revise their manuscript and make it scientifically significant enough for this journal, I can offer two different suggestions. Either they need to extend their GOSAT analysis to 5+ years and address questions such as long term trends and interannual variability of $CO_2$ fluxes. Or they need to more or less excise the GOSAT inversions and focus on the performance of GEM-MACH-GHG in simulating atmospheric $CO_2$ and its meteorological errors.

For the first choice, I would suggest questions such as:

1.Do GOSAT retrievals estimate a stronger European sink consistently over time, as first suggested by Reuter et al (2014) with SCIAMACHY and a single year of GOSAT data?

2.Do GOSAT retrievals require a stronger northern hemisphere uptake consistently, as noted by Houweling et al (2015) for one year?

3.According to GOSAT, which region contributes most to the interannual variability of atmospheric $CO_2$, the Tropics or semi-arid ecosystems? This has been an ongoing

debate in the atmospheric carbon community, see e.g., Baker et al (2006), Poulter et al (2014) and Ahlström et al (2015).

4.Are there persistent differences between GOSAT and surface data inversions across multiple years?

These are just some suggestions, and I'm sure the authors can think of many such questions to address with a multi-year GOSAT inversion.

On the other hand, if the authors choose to focus on GEM-MACH-GHG, then that would make for a very interesting paper as well. The authors have already addressed some of the interesting questions that arise from using a high resolution online model for $CO_2$ transport. Some additional questions could be:

1.Are high frequency variations of $CO_2$ near the surface better represented by the higher resolution model? If yes, we could potentially move to assimilating more data from surface measurement sites in the future with online models such as GEM-MACH-GHG.

2.Can one construct a "look up table" for the baseline transport-driven errors using GEM-MACH-GHG, varying (say) by region and season? How do those errors differ between surface and total column measurements? I'm looking for something like Figure 17, but much finer grained than three zonal bands. At the very least, ocean sites, coastal sites and continental sites should be separated. Similarly, for total column measurements, ocean and land soundings should be separated.

3.If inversions were performed using errors from step 2, versus more traditional prescription of errors, how do the fluxes change?

4.Is it true that transport errors matter less in assimilating a total column than assimilating surface sites or a vertical profile? This was first suggested by Rayner & O'Brien (2001), but to my knowledge never explicitly demonstrated. The crucial thing to compare here would be the size of the transport error and the size of the flux signal, since

that is small as well in the total column.

5.Is there any covariance between transport error and CO2 variation, especially along weather fronts? This has also been a topic of much debate, especially whether assimilating high frequency CO2 measurements can improve weather forecasts (Engelen et al, 2001), and whether CO2 inversions need to assimilate met observations.

Again, these are just some suggestions, and I'm sure the authors could come up with an interesting set of questions relevant to the atmospheric CO2 community that they could answer with GEM-MACH-GHG.

Other comments

======================

1.The prior fluxes in Figure 5 look strange. It is rare for me to see a terrestrial flux prior that is positive in the annual aggregate over North America, and Boreal and Temperate Eurasia. Where do those priors come from?

2.In Figure 7, I would prefer to see time averages, say over a week or month, instead of snapshots. Snapshots often display misleading variations that do not matter for what the authors are considering. This comment only holds, of course, if an equivalent of Figure 7 still exists in the revised manuscript.

3.I was surprised to see no data providers as co-authors in an inverse modeling paper. It is usual in this field to offer co-authorship to data providers, which they may or may not accept. In fact the ObsPack fair use policy explicitly states:

"Your use of this data product implies an agreement to contact each contributing laboratory to discuss the nature of the work and the appropriate level of acknowledgment. If this product is essential to the work, or if an important result or conclusion depends on this product, co-authorship may be appropriate. This should be discussed with each data provider at an early stage in the work. Contacting the data providers is not optional; if you use this data product, you must contact the data providers."

Were the data providers contacted, at the very least to let them know that an inversion study using their data was about to be submitted? If not, that is a significant oversight that needs to be corrected.

4.I have a problem with the terminology "flux signal", even though the authors made the explicit caveat that this "signal" by definition depends on the inverse model and the prior. The term "flux signal" makes it sound like it's an inherent property of the observations, which it is not. I would recommend using a different term, such as "CO2 adjustment" or "mole fraction update".

References

======================

A. Ahlström, M. R. Raupach, G. Schurgers, B. Smith, A. Arneth, M. Jung, M. Reichstein, J. G. Canadell, P. Friedlingstein, A. K. Jain, E. Kato, B. Poulter, S. Sitch, B. D. Stocker, N. Viovy, Y. P. Wang, A. Wiltshire, S. Zaehle, and N. Zeng, "The dominant role of semi-arid ecosystems in the trend and variability of the land CO2 sink," Science (80-. )., vol. 348, no. 6237, 2015.

D. F. Baker, R. M. Law, K. R. Gurney, P. Rayner, P. Peylin, A. S. Denning, P. Bousquet, L. Bruhwiler, Y.-H. Chen, P. Ciais, I. Y. Fung, M. Heimann, J. John, T. Maki, S. Maksyutov, K. Masarie, M. Prather, B. Pak, S. Taguchi, and Z. Zhu, "TransCom 3 inversion intercomparison: Impact of transport model errors on the interannual variability of regional CO2 fluxes, 1988-2003," Glob. Biogeochem. Cycles, vol. 20, no. 1, p. GB1002, Jan. 2006.

S. Basu, S. Guerlet, A. Butz, S. Houweling, O. Hasekamp, I. Aben, P. Krummel, P. Steele, R. Langenfelds, M. Torn, S. Biraud, B. Stephens, A. Andrews, and D. Worthy, "Global CO2 fluxes estimated from GOSAT retrievals of total column CO2," Atmos. Chem. Phys., vol. 13, pp. 8695–8717, 2013.

F. Chevallier, P. I. Palmer, L. Feng, H. Boesch, C. W. O'Dell, and P. Bousquet, "Toward

robust and consistent regional CO2 flux estimates from in situ and spaceborne measurements of atmospheric CO2," Geophys. Res. Lett., vol. 41, no. 3, pp. 1065–1070, 2014.

R. J. Engelen, G. L. Stephens, and A. S. Denning, "The effect of CO2 variability on the retrieval of atmospheric temperatures," Geophys. Res. Lett., vol. 28, no. 17, pp. 3259–3262, 2001.

S. Houweling, D. Baker, S. Basu, H. Boesch, A. Butz, F. Chevallier, F. Deng, E. J. Dlugokencky, L. Feng, A. Ganshin, O. Hasekamp, D. Jones, S. Maksyutov, J. Marshall, T. Oda, C. W. O'Dell, S. Oshchepkov, P. I. Palmer, P. Peylin, Z. Poussi, F. Reum, H. Takagi, Y. Yoshida, and R. Zhuravlev, "An intercomparison of inverse models for estimating sources and sinks of CO2 using GOSAT measurements," J. Geophys. Res. Atmos., vol. 120, no. 10, pp. 5253–5266, 2015.

S. M. Polavarapu, M. Neish, M. Tanguay, C. Girard, J. de Grandpré, K. Semeniuk, S. Gravel, S. Ren, S. Roche, D. Chan, and K. Strong, "Greenhouse gas simulations with a coupled meteorological and transport model: the predictability of CO2," Atmos. Chem. Phys., vol. 16, no. 18, pp. 12005–12038, 2016.

B. Poulter, D. Frank, P. Ciais, R. B. Myneni, N. Andela, J. Bi, G. Broquet, J. G. Canadell, F. Chevallier, Y. Y. Liu, S. W. Running, S. Sitch, and G. R. van der Werf, "Contribution of semi-arid ecosystems to interannual variability of the global carbon cycle.," Nature, vol. 509, no. 7502, pp. 600–3, May 2014.

P. J. Rayner and D. M. O'Brien, "The utility of remotely sensed CO2 concentration data in surface source inversions," Geophys. Res. Lett., vol. 28, no. 1, pp. 175–178, 2001.

M. Reuter, M. Buchwitz, M. Hilker, J. Heymann, O. Schneising, D. Pillai, H. Bovensmann, J. P. Burrows, H. Bösch, R. Parker, A. Butz, O. Hasekamp, C. W. O'Dell, Y. Yoshida, C. Gerbig, T. Nehrkorn, N. M. Deutscher, T. Warneke, J. Notholt, F. Hase, R. Kivi, R. Sussmann, T. Machida, H. Matsueda, and Y. Sawa, "Satellite-inferred Euro-

pean carbon sink larger than expected," Atmos. Chem. Phys., vol. 14, no. 24, pp. 13739–13753, 2014.
* * *

---

## Author Comment (AC1) · 1 Jun 2018

**Response to Reviewer 1**

Original text is in black. Our responses are in blue text. References to the revised text take the following format: RP78,L54-55 (Revised Page 78, Lines 54-55). The references are to the manuscript with changes tracked.

This paper compares the atmospheric distributions of CO2 resulting from two sets of optimized fluxes derived from GEOS-Chem using different observing systems based on in situ data and GOSAT data respectively. The results show the differences in the optimized fluxes and how their correction is transported in the atmosphere. An evaluation of the seasonal cycle and inter-hemispheric gradient is also provided. Finally, the zonal variability of the flux correction signal at different vertical levels (boundary layer, free troposphere and stratosphere) is also explored. The differences between the two sets of posterior fluxes and their atmospheric distributions highlight problems associated with spatial and temporal coverage of observing systems and their ability to constrain the surface CO2 fluxes at different temporal and spatial scales. Overall, the results point to the conclusion that the in situ observations do a better job at constraining the fluxes at global and annual time scales, leading to smaller biases in their fit with independent observations. While GOSAT data is able to better capture the seasonal cycle at northern extratropical sites. The paper is well written and well structured. However, I have some concerns on the use of atmospheric differences associated with flux correction patterns to draw conclusions on the potential representation of zonally asymmetric patterns by different observing systems. It is not possible to say that GOSAT is (potentially) better at constraining the zonal patterns without substantiating this with an assessement of the errors in zonal variability based on independent observations (e.g. zonal gradients using TCCON or in situ data). The analysis of the seasonal cycle could also be improved by looking at the seasonal amplitude and phase, instead of just providing seasonal biases which is too qualitative in my opinion. The results and conclusions would also be more robust if more than just one year and a half of data was used.

**Response:** The question raised by the Reviewer here and in the second and third bullets under General comments is an important one. The question ultimately concerns our approach of drawing conclusions from comparing posterior atmosphere adjustments due to flux increments with posterior atmospheric adjustments due to uncertain meteorology. Because it is such a different approach from any other in the literature, it should have been better explained. Therefore, we are grateful for the opportunity to clarify this. In the revised manuscript, we add a new section which mathematically describes the posterior atmospheric adjustment (referred to as the "flux signal" in the original manuscript) and shows that it comprises components due to flux adjustments, initial state adjustments and meteorological uncertainty. Clearly, if the posterior atmospheric adjustment due to flux increments is not the dominant term (for certain spatial or temporal scales), then caution must be exercised when utilising the retrieved fluxes on those scales. Thus we introduce a new diagnostic for retrieved fluxes and we use this to show that assimilating GOSAT observations gives zonal standard deviations that exceed those associated with meteorological uncertainties. We also agree (and we noted this in several places in the original manuscript) that the conclusion of whether GOSAT is better at constraining final spatial scales must rely on the availability of dense network of independent observations. Such a network does not currently exist. The TCCON and in situ surface networks are not sufficiently dense. But, but we can still point to the *potential* ability of GOSAT to see finer spatial scales, using our diagnostic since spatial scales produced by in situ-constrained fluxes do not exceed those produced by uncertain meteorology. In general, larger values of zonal standard deviations means more spatial structure is seen, but *more* does not mean *better*. However, if the threshold (of posterior atmospheric adjustments due to uncertain meteorology) is not exceeded, the spatial structure seen should not be trusted since a perturbed but equally valid wind field would give different spatial structures. Finally, it is always better to have longer runs but we all balance this desire with what is computationally feasible. Because of the systematic and consistent differences between the posterior atmospheric adjustments obtained with the two different posterior fluxes over the course of our experiments, we believe that our conclusions concerning the global/annual time scale and seasonal cycle are robust. We also note that our diagnostic is model dependent so there is inherent difficulty in generalizing results but our use of two very different models yielded the same patterns for atmospheric flux adjustments. To be sure, the data assimilation problem refers to a "best" estimate projected onto a given model's basis and is thus diagnostics of data assimilation systems are inherently model dependent. We had emphasized this in our original conclusions section along with the need for independent confirmation with other models. We further clarify this point in the revised manuscript.

In summary, we appreciate the thoughtful and thought-provoking comments of the Reviewer and we feel that in the course of addressing the questions raised by the Reviewer, the manuscript has been significantly improved.

GENERAL COMMENTS

- The use of CO2 flux signal to denote the cumulative impact of the flux corrections/adjustments in the atmosphere is a bit misleading. A flux signal gives the impression that it is associated to a process or phenomenon, while here it just reflects a correction (or analysis increment) which depends on the specific model, prior flux and observation used. I would think that using the term 'posterior atmospheric adjustment' would be a better term to describe the difference between posterior and prior atmospheric distributions of CO2 or alternatively 'flux correction signal'.

  **Response**: We struggled with to find a good name for "flux signal" and had considered "analysis increment". However, since it was the flux that was incremented not the concentrations, we thought that term would not work. We also agree that the term "flux signal" is not entirely appropriate as it suggests a physical phenomenon. We like the reviewer's suggestion of "Posterior Atmospheric Adjustment (PAA)" and have replaced the term "flux signal" with PAA in the revised manuscript.

- Spatial variability of flux correction signal in the atmosphere does not necessarily translate in better provision of information by observations nor an improvement in spatial/regional patterns. If the observations are very noisy (e.g. GOSAT has larger errors than the in situ observations assimilated in flux inversion systems) or observations are not homogeneously distributed (e.g. many more data over land than sea as it is the case in northern extratropical regions) then the flux corrections can create artifacts in the zonal variability which increase the zonal variability but are nevertheless not realistic.

  **Response**: We agree with this point: zonal structures in the flux signal do not necessarily imply that the inversion is capturing the true zonal structures. As the reviewer points out, biases in the observations or artifacts due to the uneven spatiotemporal distribution of the observations could plausibly result in zonal structures in the posterior fluxes. Therefore, since we do not validate the zonal structures introduced by the inversion, we cannot conclude that their presence implies that the GOSAT inversion is better capturing the true zonal structures. Although we do point this out in the previous manuscript (Page 21 lines 12-18, Page 22, line 23-28), there were also some places where we erroneously made this connection and have removed these statements. In addition, we have expanded the text discussing the possibility of spurious results. Here are some details. Newly added text is in red, deleted text is struck-out:

  Page 1, line 9 (Key point 1 was revised to):  Inversions constrained with GOSAT data introduce zonally asymmetric structures in posterior atmospheric adjustments that exceed those due to uncertain meteorology in the lower troposphere year round in the tropics and outside the boreal winter in the northern extratropics. (RP1,L10-13)

  Page 19, line 23:  (RP25,L20-21)

  Page 20, line 18-25: "In the lower troposphere, zonal asymmetry in GOSAT flux signals exceeds that arising from wind field uncertainty except in November, December and January (Figure 16a). However, for in situ data, the zonal structure can only be trusted in boreal summer (June, July and August). Thus the satellite data are potentially able to retrieve fluxes on finer spatial scales than are in situ data through most of the year, but it is important to note that more spatial structure does not mean better spatial structure. Validation of spatial structures in posterior distributions needs to be made against a dense network of independent observations in order to determine if the increased spatial variation is correct. Given the difference in observation densities ( Figures 1 and 2), this result is not surprising. The lack of ability of in situ data to produce zonal asymmetry in flux signals that are larger than those arising from uncertainty in wind fields outside of boreal summer may indicate why it has been difficult for flux inversions to

regionally attribute sources with this observation network (e.g. Gurney et al., 2002, Peters et al., 2010, Bruhwiler et al., 2011, Peylin et al., 2013)." (RP26,L19-22)

Page 22, lines 2-6: "Zonal standard deviations of the  PAAF (which reveal spatial structures in the zonal direction) are much larger when GOSAT-informed posteriors are used (in the northern extratropics outside of boreal winter and in the tropics throughout the year) (Figure 16, 17).  (RP28,L22-23)

Page 22, line 18-30: "In situ observations were found to generate zonal standard deviations larger than this minimum level only in boreal summer whereas GOSAT data exceeded this threshold through most of the year (Figure 16, 17). This potential for retrieving finer spatial scales with GOSAT sampling relative to the in situ network makes sense given the density of GOSAT observations (Figure 2) and is consistent with the prediction of Takagi et al. (2014) or Deng et al. (2016). Moreover, the ability to retrieve zonal structure is evident throughout the year in the tropics and in all seasons except boreal winter in the northern extratropics is rather encouraging. However, verifying such finer scales will be challenging given the limited spatial coverage of validating measurements from TCCON or aircraft platforms and temporal and spatial scales resolved may depend on the characteristics of the flux inversion system. Indeed, the current dispute over the enhanced European sinks obtained with GOSAT data (Feng et al., 2016; Reuter et al., 2014; Houweling et al., 2015) indicates that the finer spatial scales retrieved are not necessarily correct and are difficult to validate. It is plausible that spurious zonal structures in the PAAF could be introduced by spatially varying biases in the observations or uneven spatial coverage. However, there is also evidence supporting the ability of space-based observations to recover zonal asymmetries in the $CO_2$ fields. Liu et al. (2017) use observations from GOSAT and OCO-2 to isolate tropical flux anomalies between continents during the 2015-2016 El Niño event, while Chatterjee et al. (2017) found that zonal asymmetries in $XCO_2$ anomalies could be isolated during the same El Niño event. Furthermore, the fact that the spatial structure seen in  PAAFs obtained with in situ data surpassed the minimum uncertainty level only in boreal summer implies that regional attribution of fluxes may be challenging with the in situ observation network alone when the inversion integrates signals over many seasons." (RP29,L15-19)

Page 23, line-14-15: As noted in response below.

Newly added citations:

Liu et al. (2017):  J. Liu et al., Contrasting carbon cycle responses of the tropical continents to the 2015–2016 El Niño. Science 358, eaam5690 (2017).

Chatterjee et al. (2017):  A. Chatterjee, Influence of El Niño on atmospheric CO2 over the tropical Pacific Ocean: Findings from NASA's OCO-2 mission. Science 358, eaam5776 (2017)

- The paper would benefit from a better quantification of error reduction at different scales based on TCCON and in situ observations which could be presented in tabular format.
  **Response:**  Given that there are only 14 TCCON stations available worldwide in our period of study, it is not possible to assess spatial scales with this network.   The surface network is more plentiful, but with only around 40 stations reporting hourly, this too will not be enough to compute zonal asymmetry since most of the sites are clustered in Europe and North America.  This is why we had stressed (abstract, last line; p.21, lines 14-15; p.22, line 5)  that a dense network of independent observations is needed to verify an improvement in spatial scales achieved by any system, but that such a network does not yet exist.

- The fact that the minimal level of uncertainty in the zonal variability associated with imperfect knowledge of winds is around 0.5 ppm and the global zonal variability of flux corrections is of similar magnitude does not make the posterior zonal flux correction pattern is unreliable. The objective of the flux inversion systems is to reduce the uncertainty of the posterior fluxes and the flux corrections on their own do not necessarily reflect the uncertainty reduction. The posterior zonal patterns should be assessed with independent observations and their standard error compared to the minimal level of uncertainty associated with transport.

**Response**: The reviewer suggests that comparing posterior atmospheric adjustments due to fluxes with adjustments due to imperfect knowledge of winds is not the appropriate comparison. We disagree. The posterior atmospheric adjustment due to flux increments can and should be compared to the atmospheric adjustment due to uncertain meteorology and this can lead to new insights on the information provided by different observing systems. This becomes clear when we look at the mathematical definitions. Let us first define an atmospheric transport model ($T$) that integrates an initial state for $CO_2$ ($c_0$ ), a set of surface fluxes ($s_{0,n-1}$) and a set of meteorological fields ($x_{0,n-1}$) to yield a $CO_2$ distribution at time step $n$:

$$c_n = T(x_{0,n-1}, c_0, s_{0,n-1}). \qquad (1)$$

Subscripts refer to time steps and the model integration starts at time step 0 and yields a final $CO_2$ state at time step $n$. The posterior atmospheric adjustment (using the Reviewers' suggested name above) is simply the difference between the constituent distribution obtained with posterior fluxes and that obtained with prior fluxes:

$$\Delta c_n = T(x_{0,n-1}^a, c_0^a, s_{0,n-1}^a) - T(x_{0,n-1}^b, c_0^b, s_{0,n-1}^b) \quad (2)$$

The superscripts $a$ in (2) can be viewed as the "after adjustment" values and the superscript $b$ refers to the before adjustment value. Note that (2) is a general form which allows for the adjustment of the initial concentrations ($c_0$ ), and imperfect meteorological states ($x_{0,n-1}$) at the same time that the fluxes are adjusted. Because initial concentrations are not adjusted in our GEOS-Chem flux inversions, we ignore the impact of potential variations of $c_0$ on $\Delta c_n$. However, we allow for the possibility that the meteorological states are not perfect. Thus $x_{0,n-1}^a$ is a set of meteorological states which are perturbed by realizations of meteorological analysis error (computed as explained in our supplemental section, using our operational Ensemble Kalman filter). Then (2) can be written as:

$$\Delta c_n = T(x_{0,n-1}^a, c_0^a, s_{0,n-1}^a) - T(x_{0,n-1}^a - \varepsilon_{0,n-1}, c_0^a, s_{0,n-1}^a - \Delta s_{0,n-1}) \ (3)$$

Now, expanding (3) in Taylor series reveals that to first order, the posterior atmospheric adjustment has two components

$$\Delta c_n \cong \Delta c_n^s + \Delta c_n^x \qquad (4)$$

where

$$\Delta c_n^s = T(x_{0,n-1}^a, c_0, s_{0,n-1}^a) - T(x_{0,n-1}^a, c_0, s_{0,n-1}^b) = PAAF \quad (5)$$

$$\Delta c_n^x = T(x_{0,n-1}^a, c_0, s_{0,n-1}^a) - T(x_{0,n-1}^b, c_0, s_{0,n-1}^a) = PAAM \quad (6)$$

PAAF is the component of PAA due to flux adjustments while PAAM is the component of PAA due to uncertain meteorology. PAAF is computed by integrating the transport model with a set of posterior fluxes and again with the prior fluxes but both integrations use the same set of meteorological analyses ($x_{0,n-1}^a$) and initial concentrations. However, this is only one component of the posterior flux adjustment because the meteorological analyses are not perfectly known, and we can simulate that uncertainty by perturbing the meteorological analyses with realizations of meteorological analysis error (see supplemental material). In other words, for a given set of fluxes, the meteorological fields could have been slightly different but equally valid in the context of the meteorological analysis errors. This is what PAAM defines and it is computed by integrating the model twice (with perturbed and unperturbed meteorology) for a given set of posterior fluxes and where we again use the same initial concentrations in both integrations. A novel aspect of our work is the ability to compare the component of posterior atmospheric adjustment due to flux increments (PAAF) with that due to meteorological uncertainty (PAAM). If the PAA component due to flux increments alone (PAAF) does not exceed the component due to meteorological errors (PAAM), then it is not the dominant contribution in (5) and should not be accorded great significance.

In summary, a novel aspect of our work is the ability to compute two components of PAA with our coupled meteorological and tracer transport model. This comparison provides new insights into the atmospheric adjustments arising from flux increments associated with different types of observing systems. This information is complementary to direct comparisons of posterior $CO_2$ distributions to measurements made with a single set of meteorological analyses.

In the revised manuscript, the PAA is now defined mathematically in a new section (2.3). This greatly clarifies the arguments and results presented in our manuscript.

SPECIFIC COMMENTS
- Page 7, Line 14: Isn't the uncertainty of 22% associated with NEE very low?
**Response**:  The uncertainty of 22% is applied to both GPP and Respiration because these are the quantities that are optimized in the GEOS-Chem inversion. These quantities are significantly larger than NEE.  So too are their uncertainties.

- Page 7, Line 19: I would not call GOSAT coverage "dense".
**Response**:  We changed "dense" to "more dense" since it is fair to say GOSAT coverage is more dense than the surface network. (RP9,L12)

- Page 8, Lines 2-3: "Note that ... " sentence is not clear.
**Response**:  We clarified this statement from "Note that posterior fluxes contain total fluxes from unoptimized as well as optimized fluxes."  To "Note that posterior fluxes contain the total of all optimized (GPP, Respiration, ocean, biomass burning and anthropogenic) fluxes and the small amount of un-optimized fossil fuel emissions from shipping (~0.19PgC/yr) and aviation (~0.16PgC/yr)."  (RP9,L31-32 to RP10,L1-2)

- Page 9: Please provide a quantitative estimate of standard error and bias per month/season for the surface in situ evaluation in order to assess the seasonal cycle quantitatively. When the bias is shown to be smaller, it would help to know by how much
**Response**:  This is a good suggestion.  Table 2 was added to the manuscript.  It contains the seasonally aggregated statistics (means and standard deviations) for 6 seasons used in the TCCON figures and table, for the four experiments (two models forced by posterior fluxes from GEOS-Chem inversions with GOSAT or in situ data).  This new table is indeed helpful to the discussion of Figure 4 (revised Figure 5) on page 9 (RP13-14).

- Page 10, Line 2, Page 11, Line 23: Is the flux correction signal in the atmosphere "propagated" or "transported"?
**Response**:  In the first instance, we changed "…before considering the vertical propagation of the flux signal in section 3.2." to "…before considering the vertical structure of the PAAF in section 3.2."  (RP15,L6-7). In the second instance, we changed "…flux signal then propagates vertically…" to "flux perturbation is then vertically transported…"  (RP17,L4-5)

- Page 12, Line 16: Why is GOSAT reducing meridional gradient? From Figs 8 and 10 it looks that the meridional gradient from the GOSAT posterior fluxes is worse than that from the in situ data.
**Response**:  From our study, we cannot say why the meridional gradient is worse than that from the in situ data, but we noted here (page 3, line 19 and page 10, line 22) that other studies have seen the same thing.  We also noted in Polavarapu et al. (2016, ACP on Page 3, Line 18): "It has been suggested that the GOSAT-based inversions shift some uptake from North Africa to Europe which reduces the north-south gradient in $CO_2$ and reduces agreement with observations (Houweling et al., 2015)"

- Page 14, Lines 22-25: How do you reconcile this with the larger bias of GOSAT versus TCCON in SH?
**Response**:  It is difficult to reconcile the larger bias of GOSAT-based $CO_2$ distributions with TCCON in the southern hemisphere sites in all seasons (relative to $CO_2$ distributions informed by in situ data) (Figure 9) with the lower bias when compared to HIPPO3 data (Figure 10).  However, we can speculate that it is due to difference in specific locations sampled by the two measurement networks.  In addition, transport error does play a role because HIPPO comparisons using GEM-MACH-GHG are consistent with the TCCON results (Figure S12).  Note that comparisons to independent surface (continuous) sites in the southern hemisphere also show a larger bias for GOSAT-based posteriors even with GEOS-Chem (e.g. at South Pole in new Table 2).  For example, in Figure R1, we show monthly statistics from Cape Matatula for GEOS-Chem integrations (top panel) and GEM-MACH-GHG integrations (lower panel).  This is also consistent with the TCCON results of Figures 8 and S11.  Because there are some results that we cannot reconcile, we made conclusions based only on the results that are consistent.  What is consistent is that lower $CO_2$ values are produced with in situ-constrained fluxes in both comparisons and both models and for long (12 or 18 month) time averages, $CO_2$ fields forced by in situ-based posteriors agree better with independent measurements.

[Figure]

**Figure R1:** Comparison of model and in situ hourly observations at NOAA's Cape Matatula site with GEOS-Chem (top panel) and GEM-MACH-GHG (bottom panel). All observations (including night time) are used in statistics. The whiskers correspond to one standard deviation and the number to the right of each dot is the number of observations used in each calculation. Both GEM-MACH-GHG simulations produce more CO2 than the corresponding GEOS-Chem simulations due to transport error mismatches. For both models, in situ informed posterior distributions better match measurements.

- Page 16, Line 2: How do you explain that GOSAT produces better fit with observations in middle to upper troposphere in boreal winter?

**Response**: We can only speculate on this. We saw that in situ posteriors better match TCCON total columns at all times except boreal summer (Figs. 9, S11) and, on long time scales, they better match both TCCON (Figs. 8, S10) and NOAA aircraft profiles (Fig. S13). However, seasonal anomalies are better captured with GOSAT posteriors (Table S1) and there is seasonal variation to the agreement with the aircraft profiles (Figs. 11, S14). So, it is possible that the better fit of GOSAT posteriors to aircraft profiles is specific to the winter season, for North America. We cannot speculate on a reason for this. However, Figure 18 shows that zonal asymmetries (standard deviations) in $CO_2$ adjustments are well below those due to meteorological uncertainty (Fig. 18b,c) for the northern extratropics in boreal winter. So for the whole zonal band, zonal asymmetries (e.g. results for North America only) are not robust because they are likely sensitive to transport error.

- Page 21, Lines 11-18: The message that GOSAT observations have the potential benefit of improving the zonal structure seems to be contradicted by the results from flux inversions using GOSAT data published in Houwelling et al (2015). Therefore, the conclusion of the potential benefit of GOSAT highlighted in the abstract can be misleading.

**Response**: We did note that the "potential benefit" aspect is contradicted by the Howelling result a few lines later in the same paragraph. However, we revised the abstract to better correspond to the revised manuscript which focuses on our new diagnostic and its potential utility for understanding flux inversion results.

- Page 23, Line 15: "GOSAT better captures zonally asymmetric structure ..." should be rephrased as this has not been proven in the paper.

**Response**: The reviewer is correct. We only showed that there was more zonally asymmetric structure, not that it was better. We changed the statement to "GOSAT better captures the seasonal cycle at northern extratropical TCCON sites." (RP30,L6)

- Page 23, Lines 24-27: Note that this type of comparison has already been done by Locatelli et al. (2013, ACP) for CH4.

**Response**: What Locatelli et al. (2013, ACP) did was to use 10 different transport models with identical initial conditions and prior fluxes to simulate observations which were then used in an inversion system with a single model. Their experiment includes transport errors in the synthetic observations which are then assimilated with the reference inversion system. While this is an interesting and useful method of exploring transport error impact on flux estimation, what we suggested in the text was rather different. Instead of forcing different models to use specific initial conditions or fluxes, we simply suggest taking the results (posterior fluxes) from different inversion

models and integrating them all with a reference transport model. This will create a convolution of transport model errors from the inversion model and the reference model (as seen in our work). If transport characteristics of the reference transport model are known, relative transport errors of the flux inversion models can be inferred, to some extent. Obviously, there are limitations with this method since it is *relative* transport errors that are obtained, but since it is very easy to do (does not require a common protocol), we suggest that it is worth considering.

- Page 24, Line 10: .. seasonal correlation of "error" covariances.
**Response**: Corrected. (RP31,L2)

- Figure 4 and Page 9: It would be good to include GEOS-Chem in Fig. 4.
**Response**: We agree that because of the content of the discussion in the text, it would be good to have curves corresponding to both models on the same plot. However, there would be too many curves if all the existing ones were retained. Therefore, we have modified Figure 4 (revised Figure 5) to show the GEM-MACH-GHG and GEOS-Chem posterior $CO_2$ distributions obtained with in situ data compared to observed time series. The analogous figure for posterior $CO_2$ distributions obtained with GOSAT posterior fluxes is not shown but the newly added Table 2 (as suggested by the Reviewer in specific comment on page 9) supports the discussion concerning both types of posterior fluxes. We also now use the quality flags on the observed data and show only those not flagged as suspicious. The text on page 9 was modified accordingly. (RP13-14)

---

## Author Comment (AC2) · 1 Jun 2018

**Response to Reviewer 2**

Original text is in black. Our responses are in blue text. References to the revised text take the following format: RP78,L54-55 (Revised Page 78, Lines 54-55). The reference are for the manuscript with changes tracked.

According to my understanding, this manuscript addresses two major topics. The first is how adjustments to surface fluxes (posterior minus prior) manifest themselves in the atmosphere. This is done by performing inversions for the first two years of GOSAT data using a variational GEOS-Chem system, and propagating the posterior and prior fluxes through a transport model. Along the way, the authors perform some evaluation of their inverse results, such as comparison to TCCON and HIPPO. The second is how that manifestation varies if a different higher resolution online atmospheric transport model is used. In my opinion, the authors spend too much time on the first topic and not enough on the second, which makes the work not significant enough for a journal like Atmospheric Chemistry and Physics. If this focus were reversed, or the first topic were explored further (explained below), it would make for a much more interesting and scientifically significant paper. The authors perform inversions of GOSAT and in situ data for two years, and look at the fluxes and resultant atmospheric $CO_2$ fields in the first two years of GOSAT, primarily focusing on 2010. They use a variational inversion technique using the GEOS-Chem transport model. Their conclusions are very similar to previously published literature, such as Houweling et al (2015), Basu et al (2013), Chevallier et al (2014), which they cite. In fact, a very similar (if not identical) set of inversions was already submitted by some of the co-authors to an intercomparison of GOSAT inversions published by Houweling et al (2015). As far as I can tell, there is nothing new or unique about their inversion or analysis compared to the multitude of GOSAT inversions already published for 2010, and this part of the work does not add to the body of existing knowledge about GOSAT retrievals and derived fluxes in and around 2010. GOSAT has been up for eight years now, and retrievals of column $CO_2$ from GOSAT exist for the majority of that period. I do not understand why the authors have limited their study to the first couple of years of GOSAT data. If the authors want to publish a GOSAT inversion study that would be of value to the scientific community, I would recommend performing a longer term study, such as (say) the inter-annual variability of fluxes as seen by GOSAT, or the longer term trends in atmospheric CO2 and CO2 fluxes as seen by GOSAT. The current inversion study, focused on 2010 (with some padding on either side), is of limited interest. The second thread in their work, however, is more interesting. They perform forward runs with two different models of atmospheric transport driven by the same fluxes and look at the difference in the "flux signal" in the atmosphere. The non-GEOS-Chem model is the higher resolution GEM-MACH-GHG, a fairly new addition to this community (Polavarapu et al, 2016). Not only did they transport $CO_2$ with GEM-MACH-GHG, they also perturbed the transport with analysis errors from the meteorological assimilation system, thereby simulating the impact of uncertainties in the met fields on $CO_2$ variations. They derive a "baseline" $CO_2$ variation from this error propagation, contending that variations smaller than this detected by an observing system cannot be reliably ascribed to fluxes. This, to my knowledge, is fairly unique in the tracer transport community, and provides a recipe for deriving transport errors in $CO_2$ space. Such errors can be used, e.g., if GEM-MACH-GHG or a derived offline model is used for trace gas inversions. This technique may also be valid for deriving "baseline" transport errors for an offline model if an ensemble is run for the parent model with greenhouse gases (e.g., GEOS5 for GEOS-Chem).

If the authors would like to revise their manuscript and make it scientifically significant enough for this journal, I can offer two different suggestions. Either they need to extend their GOSAT analysis to 5+ years and address questions such as long term trends and interannual variability of $CO_2$ fluxes. Or they need to more or less excise the GOSAT inversions and focus on the performance of GEM-MACH-GHG in simulating atmospheric $CO_2$ and its meteorological errors. For the first choice, I would suggest questions such as:

1. Do GOSAT retrievals estimate a stronger European sink consistently over time, as first suggested by Reuter et al (2014) with SCIAMACHY and a single year of GOSAT data?
2. Do GOSAT retrievals require a stronger northern hemisphere uptake consistently, as noted by Houweling et al (2015) for one year?
3. According to GOSAT, which region contributes most to the interannual variability of atmospheric $CO_2$, the Tropics or semi-arid ecosystems? This has been an ongoing debate in the atmospheric carbon community, see e.g., Baker et al (2006), Poulter et al (2014) and Ahlström et al (2015).
4. Are there persistent differences between GOSAT and surface data inversions across multiple years?

These are just some suggestions, and I'm sure the authors can think of many such questions to address with a multi-year GOSAT inversion. On the other hand, if the authors choose to focus on GEM-MACH-GHG, then that would make for a very interesting paper as well. The authors have already addressed some of the interesting questions that arise from using a high resolution online model for $CO_2$ transport. Some additional questions could be:

1. Are high frequency variations of $CO_2$ near the surface better represented by the higher resolution model? If yes, we could potentially move to assimilating more data from surface measurement sites in the future with online models such as GEM-MACH-GHG.
2. Can one construct a "look up table" for the baseline transport-driven errors using GEM-MACH-GHG, varying (say) by region and season? How do those errors differ between surface and total column measurements? I'm looking for something like Figure 17, but much finer grained than three zonal bands. At the very least, ocean sites, coastal sites and continental sites should be separated. Similarly, for total column measurements, ocean and land soundings should be separated.
3. If inversions were performed using errors from step 2, versus more traditional prescription of errors, how do the fluxes change?
4. Is it true that transport errors matter less in assimilating a total column than assimilating surface sites or a vertical profile? This was first suggested by Rayner & O'Brien (2001), but to my knowledge never explicitly demonstrated. The crucial thing to compare here would be the size of the transport error and the size of the flux signal, since that is small as well in the total column.
5. Is there any covariance between transport error and $CO_2$ variation, especially along weather fronts? This has also been a topic of much debate, especially whether assimilating high frequency $CO_2$ measurements can improve weather forecasts (Engelen et al, 2001), and whether $CO_2$ inversions need to assimilate met observations. Again, these are just some suggestions, and I'm sure the authors could come up with an interesting set of questions relevant to the atmospheric $CO_2$ community that they could answer with GEM-MACH-GHG.

**Response**: We are grateful to the Reviewer for the comments and suggestions. It is clear from the comments that the organization of our original manuscript led to some confusion about the focus of the manuscript. Specially, the main focus of the manuscript is on the new diagnostic associated with the posterior atmospheric $CO_2$ adjustment. We felt that it was necessary to provide a thorough evaluation of the posterior fluxes so that the reader would be able to interpret the main results of the work. As we noted in our manuscript, our fluxes are for the most part documented in the literature. Nevertheless, it would be impossible for us to proceed without describing them (at least briefly) here for two reasons. (1) Modeling and data assimilation systems are constantly changing and improving. Because of the passage of time, the inversions done here are close to but not identical to those shown in Deng et al. (2014, 2016). It is therefore important to assure the reader that our inversion results are understandable and reasonable. (2) The flux inversion estimates under study here were described in two separate papers (Deng et al. 2014, 2016) and the two inversions results were never presented together. Because the main part of the paper compares the impact of these two posterior fluxes on atmospheric distributions, it is useful to see them in a side-by-side comparison. Furthermore, we felt that it would be unreasonable to expect the reader to be familiar with both Deng et al. papers to be able to interpret the results presented in the manuscript. However, we acknowledge the Reviewer's concern and have shortened the discussion of the flux inversion results in section 3.1 to only those remarks necessary for understanding the later sections. Specifically, we stick to only a discussion of zonally averaged fluxes. Figure S6 replaces Figures 5 which was moved to the supplemental material. Figure 6 was deleted. We also now explicitly state that the reasons for very briefly describing the posterior fluxes.

Another reason we felt that it was necessary to go into detail about the evaluation of the flux inversions is because our diagnostic is based on atmospheric adjustments from a prior distribution, therefore, when we compare adjustments from multiple posterior fluxes, we cannot determine which is more "correct". However, we can infer which is "better" by comparing posterior $CO_2$ fields to measurements. Thus, the comparisons to observations is not done to validate the fluxes, but rather to inform the discussion of the posterior atmospheric adjustments resulting from the fluxes. The importance of these comparisons is recognized by Reviewer 1 since his/her comments primarily focus on improving the discussion related to validation of the posterior $CO_2$ distributions. In retrospect, the narrative in the original manuscript may have been deficient in this regard. However, in the revised manuscript, the focus of our work is squarely on the new diagnostic and its potential value for learning about flux inversion results. This involved rewriting the abstract and revising portions of the introduction and discussion sections as well as adding explanatory sentences throughout the results section. Finally, we appreciate the enthusiastic comments

from the Reviewer regarding our new approach of comparing posterior atmospheric adjustments due to fluxes with those due to meteorological uncertainty.

In summary, as a result of the Reviewer's comments, we have rewritten portions of the manuscript to improve clarity and to focus on our main points: the introduction of a new diagnostic which is useful for studying flux inversion model results.

**Other comments**

1. The prior fluxes in Figure 5 look strange. It is rare for me to see a terrestrial flux prior that is positive in the annual aggregate over North America, and Boreal and Temperate Eurasia. Where do those priors come from?

**Response**:  As mentioned in Deng et al. (2014), the priors come from the Boreal Ecosystem Productivity Simulator (BEPS) model (Chen et al. 1999) driven by NCEP reanalysis data and remotely sensed leaf area index. Deng and Chen (2012) scaled the hourly GPP and Respiration (RSP) to generate a set of GPP and RSP  that are annually neutral for each grid box as priors for atmospheric inversion modeling. The positive priors mentioned here is/are caused by adding the biomass burning emissions. The addition of biofuel and biomass burning sources to the natural fluxes was noted in the caption for Figure 5. However, this figure was moved to the supplementary section (Figure S6) in the revised manuscript.

Chen, J. M., Liu, J., Cihlar, J., and Goulden, M. L.: Daily canopy photosynthesis model through temporal and spatial scaling for remote sensing applications, Ecol. Model., 124, 99–119, 1999.

2. In Figure 7, I would prefer to see time averages, say over a week or month, instead of snapshots. Snapshots often display misleading variations that do not matter for what the authors are considering. This comment only holds, of course, if an equivalent of Figure 7 still exists in the revised manuscript.

**Response**:  In general, time averages are typically preferred for the reasons the reviewer presents.  However, in our manuscript, we need snapshots and not time averages.  Figure 7 is an encapsulation of the full time animations provided in the supplemental material.  The evolution of the flux signal reveals interesting differences and hints at transport pathways that are explored in the section on adjoint sensitivity (and Figure 12).  The time evolution of the animations (which the snapshots summarize) also corresponds to the time evolution of flux signals seen in later figures (Figures 13-18).  Furthermore, the qualitative consistency of differences between the two types of flux signals across the two years, given that these are snapshots, is additional useful information.

3. I was surprised to see no data providers as co-authors in an inverse modeling paper. It is usual in this field to offer co-authorship to data providers, which they may or may not accept. In fact the ObsPack fair use policy explicitly states:

> *"Your use of this data product implies an agreement to contact each contributing laboratory to discuss the nature of the work and the appropriate level of acknowledgment. If this product is essential to the work, or if an important result or conclusion depends on this product, co-authorship may be appropriate. This should be discussed with each data provider at an early stage in the work. Contacting the data providers is not optional; if you use this data product, you must contact the data providers."*

Were the data providers contacted, at the very least to let them know that an inversion study using their data was about to be submitted? If not, that is a significant oversight that needs to be corrected.

**Response**: We are well aware of protocols for data usage and we fully appreciate the dedication and effort (by our close colleagues and by all measurement scientists) required to make high quality measurements.  For that reason, we always try to acknowledge the expertise and effort required to make measurements and to provide them to researchers.  Here are some specific details, regarding this manuscript.

   a) TCCON.  Some of our authors are co-located with TCCON PI Kim Strong at the University of Toronto.  As with previous work, we contacted all TCCON PIs whose data appears in the tables and Figures.  We also had discussions with Kim Strong about Eureka data, but we had no requests from any TCCON PIs for co-authorship.  This is the same procedure and same result we got in our previous article (Polavarapu et al. 2016, ACP).  The references for all 14 sites were cited in the text and all PIs were thanked in the acknowledgements.

b) HIPPO. We contacted Steve Wofsy by email (21 March 2017) and received no reply. This was also the case with our previous publication. (Also, in the case of the Deng et al. (2016) manuscript, Steve Wofsy declined co-authorship.) In the absence of a reply for this manuscript, we referenced his papers and the dataset DOI and added an acknowledgement. Note that the data usage protocol found at https://www.eol.ucar.edu/system/files/HIPPO_Full_Data_Policy_lah_20170915_1.pdf requests only the following: (1) Acknowledge with references and use the DOI number, (2) Acknowledge NSF and NOAA in the acknowledgements, (3) add "HIPPO, HIAPER Pole-to-Pole Observations, National Science Foundation, NSF, NSF/NCAR Gulfstream-V (GV)" to keywords. Note that we had done all of these in the original manuscript.

c) NOAA aircraft profiles. We contacted Colm Sweeney by email (8 Nov 2017) and provided the manuscript, figures and supplemental material and received no response. In the absence of a response, we cited his publications, added an acknowledgement to him, and submitted the manuscript on 29 Dec 2017. NOAA aircraft profiles were obtained from ObsPack2013 which was acknowledged as noted in (d).

d) NOAA surface measurements. For ObsPack, the contact in the datafiles mentioned is Ken Masarie. But we heard from NOAA colleagues that Ken had retired. Therefore, we contacted Arlyn Andrews (8 Nov 2017). Again, we provided the manuscript, figures and supplemental information. We did not get a reply. In the absence of a response, we cited Conway and Tans (2012), Conway et al. (2011), Masarie et al. (2014) and the DOI (http://dx.doi.org/10.3334/OBSPACK/1001).

4. I have a problem with the terminology "flux signal", even though the authors made the explicit caveat that this "signal" by definition depends on the inverse model and the prior. The term "flux signal" makes it sound like it's an inherent property of the observations, which it is not. I would recommend using a different term, such as "$CO_2$ adjustment" or "mole fraction update".

**Response**: Reviewer 1 had exactly the same concern and suggested an alternative expression which we like: "Posterior Atmospheric Adjustment". We adopted this new terminology and its acronym (PAA) in the revised manuscript.

**References**
A. Ahlström, M. R. Raupach, G. Schurgers, B. Smith, A. Arneth, M. Jung, M. Reichstein, J. G. Canadell, P. Friedlingstein, A. K. Jain, E. Kato, B. Poulter, S. Sitch, B. D. Stocker, N. Viovy, Y. P. Wang, A. Wiltshire, S. Zaehle, and N. Zeng, "The dominant role of semi-arid ecosystems in the trend and variability of the land CO2 sink," Science (80-.)., vol. 348, no. 6237, 2015.

D. F. Baker, R. M. Law, K. R. Gurney, P. Rayner, P. Peylin, A. S. Denning, P. Bousquet, L. Bruhwiler, Y.-H. Chen, P. Ciais, I. Y. Fung, M. Heimann, J. John, T. Maki, S. Maksyutov, K. Masarie, M. Prather, B. Pak, S. Taguchi, and Z. Zhu, "TransCom 3 inversion intercomparison: Impact of transport model errors on the interannual variability of regional CO2 fluxes, 1988-2003," Glob. Biogeochem. Cycles, vol. 20, no. 1, p. GB1002, Jan. 2006.

S. Basu, S. Guerlet, A. Butz, S. Houweling, O. Hasekamp, I. Aben, P. Krummel, P. Steele, R. Langenfelds, M. Torn, S. Biraud, B. Stephens, A. Andrews, and D. Worthy, "Global CO2 fluxes estimated from GOSAT retrievals of total column CO2," Atmos. Chem. Phys., vol. 13, pp. 8695–8717, 2013.

F. Chevallier, P. I. Palmer, L. Feng, H. Boesch, C. W. O'Dell, and P. Bousquet, "Toward robust and consistent regional CO2 flux estimates from in situ and spaceborne measurements of atmospheric CO2," Geophys. Res. Lett., vol. 41, no. 3, pp. 1065–1070, 2014.

R. J. Engelen, G. L. Stephens, and A. S. Denning, "The effect of CO2 variability on the retrieval of atmospheric temperatures," Geophys. Res. Lett., vol. 28, no. 17, pp. 3259–3262, 2001.

S. Houweling, D. Baker, S. Basu, H. Boesch, A. Butz, F. Chevallier, F. Deng, E. J. Dlugokencky, L. Feng, A. Ganshin, O. Hasekamp, D. Jones, S. Maksyutov, J. Marshall, T. Oda, C. W. O'Dell, S. Oshchepkov, P. I. Palmer, P. Peylin, Z. Poussi, F. Reum, H. Takagi, Y. Yoshida, and R. Zhuravlev, "An intercomparison of inverse models for estimating sources and sinks of CO2 using GOSAT measurements," J. Geophys. Res. Atmos., vol. 120, no. 10, pp. 5253–5266, 2015.

S. M. Polavarapu, M. Neish, M. Tanguay, C. Girard, J. de Grandpré, K. Semeniuk, S. Gravel, S. Ren, S. Roche, D. Chan, and K. Strong, "Greenhouse gas simulations with a coupled meteorological and transport model: the predictability of CO2," Atmos. Chem. Phys., vol. 16, no. 18, pp. 12005–12038, 2016.

B. Poulter, D. Frank, P. Ciais, R. B. Myneni, N. Andela, J. Bi, G. Broquet, J. G. Canadell, F. Chevallier, Y. Y. Liu, S. W. Running, S. Sitch, and G. R. van der Werf, "Contribution of semi-arid ecosystems to interannual variability of the global carbon cycle.," Nature, vol. 509, no. 7502, pp. 600–3, May 2014.

P. J. Rayner and D. M. O'Brien, "The utility of remotely sensed CO2 concentration data in surface source inversions," Geophys. Res. Lett., vol. 28, no. 1, pp. 175–178, 2001.

M. Reuter, M. Buchwitz, M. Hilker, J. Heymann, O. Schneising, D. Pillai, H. Bovensmann, J. P. Burrows, H. Bösch, R. Parker, A. Butz, O. Hasekamp, C. W. O'Dell, Y. Yoshida, C. Gerbig, T. Nehrkorn, N. M. Deutscher, T. Warneke, J. Notholt, F. Hase, R. Kivi, R. Sussmann, T. Machida, H. Matsueda, and Y. Sawa, "Satellite-inferred European carbon sink larger than expected," Atmos. Chem. Phys., vol. 14, no. 24, pp. 13739–13753, 2014.

---

## Editor Decision (ED1)

Dear Saroja,

Thank-you for the work that you and your co-authors have undertaken to address the reviewers comments on your manuscript. Overall I think you have responded sufficiently to the comments from reviewer 1 and your new section 2.3 is helpful in clarifying the methodology. However this clarification has highlighted a difficulty for me in the comparison between PAAF and PAAM, which I think may also be relevant to your response to reviewer 2.

My concern is that PAAF is a relative measure since it is dependent on your prior flux. In this regard the comparison of PAAF across inversions (using the same prior) is reasonable but I think you need to be more careful in how you describe the comparison between PAAF and PAAM. For example, if you ran your inversion with an unrealistic prior (e.g. no seasonal cycle) then it seems plausible to me that PAAF would easily be larger than PAAM everywhere for all seasons whereas if you started with a prior that closely matched the observations then PAAF would be small and you would conclude that uncertainty from meteorology dominates. The relative nature of PAAF is not always clear in your manuscript. For example, on p27, line 2 you write 'the zonal structure in the tropics is trustable only in …' where perhaps this should be written 'the zonal structure is … differentiable from the prior only in …' Likewise, is it clear that the GOSAT inversion adds zonally asymmetric structure as you suggest, or only that it changes the zonal asymmetry relative to the prior? With these examples in mind, please could you check through your manuscript again to ensure that your text reflects the relative nature of PAAF and that the conclusions you draw from your PAAF/PAAM comparison are justified.

Reviewer 2 provided you with two options for making the manuscript more significant. You have taken their 2nd option, focussing on the PAAF/PAAM analysis, but my reading of their review is that they would like to see more analysis and discussion of PAAM particularly. They suggest 5 additional questions that you could address, but you do not appear to have taken up any of these suggestions or provided a response to the reviewer as to the value or feasibility of their suggestions. Please can you respond to these 5 suggestions and consider whether one or two could be helpfully added to your paper. Personally, I think suggestion 2 could provide some interesting insights into transport error that might be helpful for other inverse modellers. Given your paper is already quite long, please also consider whether any sections can be shortened or figures removed to accommodate any additional analysis. For example, I wonder whether Figures 13-15 are all needed to illustrate the main points you make in Sec 3.2.4, and likewise for Figures 16-18/Sec 3.2.5. Alternatively perhaps the comparison to TCCON, HIPPO and the aircraft profiles could be shortened.

My re-reading of the manuscript also identified some possible technical corrections.

P17, line 18. 'uptake' needed after '2.5 Pg C per year'?

P18, line 6. Replace 'relatively lower' with 'less negative'?

P22, line 22. 'global mean' instead of 'globally averaged zonal mean'?

Please feel free to contact me by email if any of these comments are not clear.

Regards,

Rachel Law, rachel.law@csiro.au

---

## Author Response (AR2)

**Response to Editor's comments**

After the revision of our manuscript (which was submitted on June 1, 2018) we received comments from the Editor with suggestions for minor revisions on June 19, 2018. Our responses to some of these suggestions are given below in section 1. Clarification of the remaining suggestions was requested by the authors via direct email (on June 22, 2018) with the Editor (Rachel Law). The authors' email contained explicit responses to suggestions from Reviewer 2 for further studies and are attached. Therefore, those responses will not be repeated here. However, the Editor's response to the request for clarification provided further guidance. Our responses to those suggestions are included in section 2, below.

Original text is in black. Our responses are in blue text. Page and line number references are to the most recent manuscript version with changes tracked. In addition to the suggested changes below, we also improved the quality of the schematic diagram in Figure 4.

1) **acp-2017-1235-comments-to-author.pdf (July 19, 2019):**

Thank-you for the work that you and your co-authors have undertaken to address the reviewers comments on your manuscript. Overall I think you have responded sufficiently to the comments from reviewer 1 and your new section 2.3 is helpful in clarifying the methodology. However this clarification has highlighted a difficulty for me in the comparison between PAAF and PAAM, which I think may also be relevant to your response to reviewer 2.

My concern is that PAAF is a relative measure since it is dependent on your prior flux. In this regard the comparison of PAAF across inversions (using the same prior) is reasonable but I think you need to be more careful in how you describe the comparison between PAAF and PAAM. For example, if you ran your inversion with an unrealistic prior (e.g. no seasonal cycle) then it seems plausible to me that PAAF would easily be larger than PAAM everywhere for all seasons whereas if you started with a prior that closely matched the observations then PAAF would be small and you would conclude that uncertainty from meteorology dominates. The relative nature of PAAF is not always clear in your manuscript. For example, on p27, line 2 you write 'the zonal structure in the tropics is trustable only in …' where perhaps this should be written 'the zonal structure is … differentiable from the prior only in …' Likewise, is it clear that the GOSAT inversion adds zonally asymmetric structure as you suggest, or only that it changes the zonal asymmetry relative to the prior? With these examples in mind, please could you check through your manuscript again to ensure that your text reflects the relative nature of PAAF and that the conclusions you draw from your PAAF/PAAM comparison are justified.

**Response**: We agree with the comments and we have gone through the manuscript to identify statements which should be better qualified. In particular, statements were added to the abstract (p.2), Introduction (p.4), the end of section 2 (p.14), and the discussion (p.29).

Reviewer 2 provided you with two options for making the manuscript more significant. You have taken their 2nd option, focussing on the PAAF/PAAM analysis, but my reading of their review is that they would like to see more analysis and discussion of PAAM particularly. They suggest 5 additional questions that you could address, but you do not appear to have taken up any of these suggestions or provided a response to the reviewer as to the value or feasibility of their suggestions. Please can you respond to these 5 suggestions and consider whether one or two could be helpfully added to your paper. Personally, I think suggestion 2 could provide some interesting insights into transport error that might be helpful for other inverse modellers. Given your paper is already quite long, please also consider whether any sections can be shortened or figures removed to accommodate any additional analysis. For example, I wonder whether Figures 13-15 are all needed to illustrate the main points you make in Sec 3.2.4, and likewise for Figures 16-18/Sec 3.2.5. Alternatively perhaps the comparison to TCCON, HIPPO and the aircraft profiles could be shortened.

**Response**: Our request for clarification (sent via email to Rachel Law on June 22, 2018) contains responses to these suggestions. Since that letter is attached, the responses will not be repeated here.

My re-reading of the manuscript also identified some possible technical corrections.

P17, line 18. 'uptake' needed after '2.5 Pg C per year'?

**Response**:  Corrected (page 16, line 20).

P18, line 6. Replace 'relatively lower' with 'less negative'?

**Response**:  This is a good suggestion.  Corrected (page 17, line 9).

P22, line 22. 'global mean' instead of 'globally averaged zonal mean'?

**Response**:  Corrected (page 21, line 22).

2)  **Email from Editor (Rachel Law) received on July 28, 2018** (subject: Guidance on acp-2017-1235 revisions):

Thank-you for your continued work to improve the manuscript. I suggest the following:

1.      As you indicate in the first paragraph of your reply, please go ahead and review and revise the version to be clear about the relative nature of the PAAF diagnostic. In this regard, I wonder whether the Fig 1 that you've included in your reply is useful. It could perhaps be included at the end of section 2.4 after you've described how PAAF and PAAM are calculated. It would become an example from which you could note such points as the relative nature of PAAF as shown by PAAF increasing in time because of the prior choice, the seasonal nature of PAAM for this site, and that for this paper you are going to focus mostly on PAAF/PAAM at larger scales (zonal mean and variability etc).

**Response**:  We added the figure comparing time series of PAAF and PAAM at Sable Island to the revised manuscript as Figure 6.  A new paragraph describing this figure was added to section 2.4 on page 14.

2.      Your replies to the reviewer 2 suggestions are reasonable and will now become part of the record for the paper so I think that is adequate. I accept that most would require substantially more work that isn't appropriate for this paper. You might want to consider incorporating some of your responses into an additional paragraph for Section 4 indicating some of the future work that you are planning (or already undertaking) out of this study and making use of these diagnostics.

**Response**:  We added new text at the end of the manuscript (p.30) discussing future work.  We also noted when discussing the new Figure 6 (p.14) that the application of the diagnostic at individual sites is the subject of ongoing investigation.

June 22, 2018

Dear Rachel,

Thank you for your feedback on our revised manuscript. We agree with your suggestion to carefully review the revised version in light of the relative nature of the measure. Although we state this point a few times, there are instances (as you aptly noted) where the conclusions need better qualification. This independently became apparent to me when I recently presented this work. Therefore, I am very glad for the opportunity to better stress that the diagnostic is always specific to given assimilation system and prior flux but it is very useful for exactly that reason.

I apologize for not explicitly responding to the suggestions of Reviewer 2 for more analysis. While these are all great suggestions, they are all significant undertakings that could follow on from this initial analysis. However, we feel that we cannot do the follow-on studies without first introducing the main approach, which is the focus of this paper. It is possible that Reviewer 2 did not fully appreciate what was novel about our work. We had hoped that with the revised version, and the emphasis on the value of the new diagnostic, that it would have become clearer. Reviewer 2 did say "these are just some suggestions, and I'm sure the authors could come up with an interesting set of questions relevant to the atmospheric $CO_2$ community that they could answer with GEM-MACH-GHG" so we felt that our revised manuscript did just that by clarifying our new diagnostic. Our approach to understanding flux increments has not previously been presented in the literature for GHGs so we feel that it is itself a significant contribution. We also spent considerable effort to make the argument a cohesive whole. As you noted, the manuscript is already long (we have seven sections and we provide lengthy supplemental information including animations) but we do not believe that any section could be cut or reduced without affecting the cohesion of the whole article. Moreover, we feel the manuscript would be less comprehensible if parts were cut out in order to take the article in a new direction. The most feasible of the 5 proposed extensions suggested by Reviewer 2 is the second one (as you had noted). In fact, we would like to take up this suggestion but it will require a lot more work and might possibly become a GRL paper someday. Moreover, the proposed extension (suggestion 2 below) is a bit distinct from the present study in that the impact of meteorological uncertainty for observations is considered, rather than the impact of meteorological uncertainty over a range of scales, which is the main topic of our paper. Below we give explicit responses to each of the 5 proposed extensions of our present work in blue text.

1. Are high frequency variations of $CO_2$ near the surface better represented by the higher resolution model? If yes, we could potentially move to assimilating more data from surface measurement sites in the future with online models such as GEM-MACH-GHG.
   Ans: This is a great question, but not an easy one to answer. I have a post-doc (Dr. Jinwoong Kim) who has spent the past year developing a regional version (with 10 km grid) of our global (90 km grid) GEM-MACH-GHG model. He has been trying to answer exactly this question. It is easy to demonstrate that the meteorology is improved in the regional model but it is hard to show that the $CO_2$ is better in the regional model. With identical coarse resolution fluxes to the global model, the regional model $CO_2$ is better only where regional predictability is expected to be systematically better: in regions of complex terrain. He is now considering the benefit of using high resolution (ODIAC) fluxes. The high resolution fluxes can produce larger amplitude diurnal cycles, but this is not necessarily correct in terms of agreement with observations. Therefore, we are unable to prove at this point that the regional model is better at simulating GHGs than the global model, though it should be, theoretically. This is a separate, on-going project which may or may not lead to a publication on this topic.

2. Can one construct a "look up table" for the baseline transport-driven errors using GEM-MACH-GHG, varying (say) by region and season? How do those errors differ between surface and total column measurements? I'm looking for something like Figure 17, but much finer grained than three zonal bands. At the very least, ocean sites, coastal sites and continental sites should be separated. Similarly, for total column measurements, ocean and land soundings should be separated.
   Ans: In this work, we don't directly compute transport errors. To do that, we would need a full Kalman Filter. Note that we are indeed developing a coupled meteorological/tracer/flux ensemble Kalman Filter but it is not ready yet. (More details on the EnKF work are given in the answer to point 5.) However, we could plot and compare PAAF and PAAM (which reflects one component of transport error) at observation sites. We show such a comparison at Sable Island in Figure 1 below. Note that PAAF contains a trend, seasonal cycle and synoptic scale variations whereas PAAM has mainly synoptic scale variations. Monthly statistics might remove much of the trend and seasonal signals from PAAF for comparison to PAAM, as an approximation to directly removing those signals (since our time series is not long enough for clean

removal of the seasonal cycle).  However, more problematic is the interpretation of such plots.  For example, PAAM variability is comparable to PAAF in summer 2010 but PAAF variability exceeds that of PAAM in summer 2009.  We would need a separate study and we would need to involve some measurement scientists in the work to better understand what we are seeing.  It would be discouraging to inverse modellers if PAAM were comparable to PAAF at most locations, so considerable diligence in this study is required.  In fact, we plan to embark on such a study with a new team and new experiments.  The results would benefit our EnKF development because we need to specify instrument plus representativeness errors for all our surface sites (and all observations, in fact) and this may provide a way to estimate this.

[Figure]

**Figure 1: Time series of PAAF and PAAM sampled at the model grid box nearest to Sable Island at the lowest model level.**

3. If inversions were performed using errors from step 2, versus more traditional prescription of errors, how do the fluxes change?
Ans:  When we have our full ensemble Kalman Filter developed, we could provide our estimates of forecast/transport error to the inverse modelling community.  Such products could be full 3D fields of ensemble standard deviations, or else, realizations of transport error as a function of time.

4. Is it true that transport errors matter less in assimilating a total column than assimilating surface sites or a vertical profile? This was first suggested by Rayner & O'Brien (2001), but to my knowledge never explicitly demonstrated. The crucial thing to compare here would be the size of the transport error and the size of the flux signal, since that is small as well in the total column.
Ans: As noted below, we are developing a coupled ensemble Kalman Filter (EnKF) and when that is done, we would be able to answer this question.  In this article, we study components of the posterior atmospheric adjustment and don't get into transport error characterization.  In Polavarapu et al. (2016, ACP) we show that transport error comprises many terms and we show that the component due to imperfect meteorology places limits on the spatial scales that can be resolved in $CO_2$.  However, the complete transport error (with contributions from flux error, initial state error, observation error, model error, etc.) was not assessed in that paper but it will be simulated once our EnKF is developed.  Then we could answer the question of the size of the transport error for surface sites versus column measurements.  In this work, we don't consider transport error since we don't have a truth, and we don't have a coupled EnKF at our disposal.  So, this is a great question, but we are still far from being able to definitely answer it.

5. Is there any covariance between transport error and $CO_2$ variation, especially along weather fronts? This has also been a topic of much debate, especially whether assimilating high frequency $CO_2$ measurements can improve weather forecasts (Engelen et al, 2001), and whether $CO_2$ inversions need to assimilate met observations.
Ans: The way to demonstrate this is to develop a coupled data assimilation for meteorology and $CO_2$.  We have been developing an ensemble Kalman filter to estimate the coupled meteorology/GHG state/GHG flux estimation problem for the past 6 years.  It is a slow, complex effort because we are using operational weather forecast models and assimilation systems (which are constantly changing, and are not backward compatible) and because we don't have a lot of resources for this project.  However, we can now produce animations of transport error due to uncertain meteorology which show nice consistency with synoptic scale transport.  (But this is only one component of transport error.  The complete transport error requires us to also account for flux and initial condition errors.)  In fact, we now simulate $CO_2$, CO and $CH_4$ simultaneously (though they don't interact) and all 3 species show similar transport patterns (Figure 2 below).  So, this is again a good question, but cannot be answered in the confines of our current manuscript.  My post-doc (Dr. Vikram Khade) who has been leading this development plans to document our system in a GMDD paper (hopefully) this year.

[Figure]

Forecast ensemble spread - 2015-01-09 09:00

**Figure 2. The ensemble spread of 64 forecasts for column mean $CO_2$ (top), $CH_4$ (middle) and (CO) bottom, on Jan. 9, 2015 at 00Z. The EnKF started on Dec. 28, 2014 00Z so errors due to loss of predictability of weather have largely saturated by this date. The forecast spread arises from imperfect meteorological fields alone. Flux errors are not accounted for in this experiment.**

In summary, all 5 suggestions of Reviewer 2 are excellent but are not directly related to the point of our manuscript. Rather, they require the completion of our coupled meteorological/tracer assimilation system. As noted above, we have been developing an EnKF for some time now and it should come to fruition soon. The EnKF will use an ensemble of forecasts to directly simulate all components of transport error. That will give us quantitative results. In the meantime, we can study the impact of a single component of transport error (that due to meteorological uncertainty) as we did in Polavarapu et al. (2016, ACP) or propose a new diagnostic tool for assimilation system development (as we do in this article). In these papers, we do not have the full transport error but we find that looking at components of transport error is also revealing.

In summary, we are grateful for the insightful and encouraging remarks from both Reviewers. We feel the manuscript has greatly benefitted from their comments. However, before making any further changes to the current version of the manuscript, we await further instructions from you.

Thank-you for your guidance,

Saroja Polavarapu
(for the co-authors: Feng Deng, Brendan Byrne, Dylan Jones and Michael Neish)

[revised manuscript text omitted]